

# Large-scale automated emission measurement of individual vehicles with point sampling

Markus Knoll[1], Martin Penz[1], Hannes Juchem[2], Christina Schmidt[2,3], Denis Pöhler[2,3], and Alexander Bergmann[1]

[1]Institute of Electrical Measurement and Sensor Systems, Graz University of Technology, Inffeldgasse 33/I, 8010 Graz, Austria
[2]Institute of Environmental Physics, Heidelberg University, INF 229, 69120 Heidelberg
[3]Airyx GmbH, Justus-von-Liebig-Str. 14, 69214 Eppelheim, Germany

**Correspondence:** Markus Knoll (markus.knoll@tugraz.at)

**Abstract.** Currently, emissions from internal combustion vehicles are not properly monitored throughout their life cycle. In particular, a small share ($< 20$ %) of poorly maintained or tampered vehicles are responsible for the majority (60-90 %) of traffic-related emissions. Remote emission sensing (RES) is a method used for screening emissions from a large number of in-use vehicles. Commercial open-path RES systems are capable of providing emission factors for many gaseous compounds, but they are less accurate and reliable for particulate matter (PM). Point sampling (PS) is an extractive RES method where a portion of the exhaust is sampled and then analyzed. So far, PS studies have been conducted predominantly on a rather small scale and have mainly analyzed heavy duty vehicles (HDV), which have high exhaust flow rates. In this work, we present a comprehensive PS system that can be used for large-scale screening of PM and gas emissions, largely independent of the vehicle type. The developed data analysis framework is capable of processing data from 1,000s of vehicles. The core of the data analysis is our peak detection algorithm (TUG-PDA), which determines and separates emissions down to a spacing of just a few seconds between vehicles. We present a detailed evaluation of the main influencing factors on PS measurements by using about 100,000 vehicle records collected from several measurement locations, mainly in urban areas. We show the capability of the emission screening by providing real-world black carbon (BC), particle number (PN) and $NO_x$ emission trends for various vehicle categories such as diesel and petrol passenger cars or HDVs. Comparisons with open-path RES and PS studies show overall good agreement and demonstrate the applicability even for the latest Euro emission standards, where current open-path RES systems reach their limits.

## 1 Introduction

Exhaust emissions from combustion-based vehicles are negatively affecting human health and our environment. Of specific interest are $NO_x$ and particulate matter (PM) emissions due to the known impact on health, environment and climate (Mannucci et al., 2015; EEA, 2017). $NO_x$ emissions remain a widespread problem, especially for diesel-powered vehicles, where tampered and defective vehicles contribute to high emission levels (Meyer et al., 2023). For PM it is well known from literature, that a small share of vehicles ($< 20$ %) contribute to the vast amount of emissions (60-90 %) due to malfunction




after-treatment systems, such as defective diesel particulate filter (DPF) (Park et al., 2011; Burtscher et al., 2019; Boveroux et al., 2019; Bainschab et al., 2020). It would be highly beneficial to human health and our environment if these high emitters could be identified and subsequently maintained in order to significantly reduce emissions. Most of current regulations are only related to type approval procedures, but do not consider deterioration (e.g., of the exhaust after-treatment system), defects which are not properly repaired or tampering that occurs in the lifetime of the vehicles (Mock and German, 2015; Bainschab et al., 2020). Particle number (PN) concentrations and black carbon (BC) are two PM metrics of particular interest. In addition to its impact on health and climate, BC is a suitable tracer for vehicles with high PM emissions (Salimbene et al., 2021; Rönkkö et al., 2023). Interest in real-world PN emissions is growing due to newly introduced regulations (Giechaskiel et al., 2021) and known health effects on the human respiratory and cardiovascular systems (Oberdörster et al., 2005; Brook et al., 2010).

Different strategies exist which try to address these issues. PN concentration measurements during periodic technical inspections (PTI) are currently implemented in several European countries like Germany, the Netherlands, Switzerland and Belgium. The PN inspections should identify malfunctioning vehicles during low-idle operation (Bainschab et al., 2020; Giechaskiel et al., 2020; Melas et al., 2021; Giechaskiel et al., 2021). A disadvantage of PTI is that they are in the best case annual, one-time measurements performed under non real driving conditions, which can potentially be circumvented by tampering or by making wrong measurements. Another approach taken for high emitter identification and the screening of real-world emissions of in-use vehicles is remote emission sensing (RES). RES is employed directly at the roadside to measure emissions from passing vehicles under real driving operating conditions (Bishop et al., 1989; Borken-Kleefeld et al., 2018). One advantage of RES is that the vehicles are measured during their normal operation, which complicates fraud. Commercially available RES systems are open-path systems that detect the light extinction of the exhaust plume at different wavelengths to measure different pollutants emitted by passing vehicles (Bishop et al., 1989; Stedman et al., 1992; Moosmüller et al., 2003; Burgard et al., 2006). These systems deliver statistically acceptable emission factors (EF) for gaseous species, but EFs are inaccurate for particulates. In particular, PM emissions of the latest Euro emission standards (Euro 6, Euro VI and beyond) are below the quantification limit of open-path RES systems (Gruening et al., 2019; Cha and Sjödin, 2022; Jerksjö et al., 2022). PN emissions cannot be accurately determined using these systems, which are currently of specific interest (de Jesus et al., 2019; Giechaskiel et al., 2021). Complementary RES concepts exist which can be applied to counteract the downsides of these systems. In plume chasing (PC), a measurement vehicle equipped with laboratory grade analyzers traces vehicles under test. Several studies (Ježek et al., 2015; Järvinen et al., 2019; Pöhler et al., 2019; Wang et al., 2020) have shown that reliable EFs can be determined by chasing the vehicle under test over a short time period. The disadvantage of PC is that it is a rather labor-intensive method which can only be applied to a small number (< 200) of vehicles per chasing vehicle and day.

Extractive point sampling (PS) is a roadside measurement technique (see Fig. 1) that can be used to capture the plumes from passing vehicles by sampling the diluted exhaust (Hansen and Rosen, 1990; Janhäll and Hallquist, 2005; Hak et al., 2009; Ban-Weiss et al., 2009). Compared to open-path RES systems the installation of the measurement setup is relatively simple. The sample is usually directly extracted at the road surface or at the roadside, as close as possible to the tailpipe of the passing vehicles. A small shelter or a van next to the sample extraction houses the instruments that analyze the captured emissions of the passing vehicles (Hak et al., 2009). PS studies have predominantly measured heavy duty vehicles (HDV) or buses by





sampling from the roadside (Hallquist et al., 2013; Watne et al., 2018; Liu et al., 2019; Zhou et al., 2020) or, by sampling from the top of tunnels or bridges for HDVs with a vertical exhaust pipe, which are common in the US (Ban-Weiss et al.,

2008, 2009, 2010; Dallmann et al., 2011, 2012; Preble et al., 2015; Bishop et al., 2015; Preble et al., 2018; Sugrue et al., 2020). Detailed analysis of fleet emissions by characteristics such as emission standard, manufacturer or vehicle age were performed mainly in PS studies measuring HDVs and buses (Dallmann et al., 2011; Bishop et al., 2015; Preble et al., 2015, 2018; Liu et al., 2019; Zhou et al., 2020). PS systems capable of large-scale emission screening independent of vehicle type are rare and have only been applied for vehicles classified by length (Wang et al., 2015, 2017) or number of axes and tires (Ban-Weiss et al.,

2008, 2010; Dallmann et al., 2013, 2014) or for gaseous compounds using sensor networks (Chu et al., 2022). To the best of our knowledge, there are only individual PS studies in which the LDV emissions of a few test vehicles were determined based on characteristics such as the Euro emission standard (Hak et al., 2009; Ježek et al., 2015). Analysis of emissions by emission standard, manufacturer or age provides more detailed information, e.g. on whether emission limits are generally being met or whether certain manufactures or vehicles stand out.

In this work, we present a comprehensive PS technique that can be used for large-scale emission screening of individual vehicles, largely independent of the vehicle type. The PS system measures different PM metrics as well as gaseous compounds from various vehicle categories such as diesel and petrol passenger cars or HDVs. We show the capability of the system by providing real-world BC, PN and $NO_x$ emission trends. The PS setup can be operated in stand-alone mode and allows emission measurements to be carried out down to a distance of just a few seconds between the vehicles. The developed data analysis

framework is capable of processing data from 1,000s of vehicles and includes all processing steps from reading raw time series data up to providing fleet emission statistics. We provide a detailed insight into the developed PS methodology by discussing the dependencies and key factors, including instrument selection criteria, measurement site selection, sample extraction, vehicle dependencies and environmental impacts. This work was conducted as part of the H2020 project city air remote emission sensing (CARES) (https://cares-project.eu/). We use the term pollutant for all measured analytes except $CO_2$. Important definitions

for RES emission calculations are described in Appendix A.

## 2 Method

### 2.1 Measurement setup

We propose a PS setup as illustrated in Fig. 1a to carry out the measurement procedure and the data post-processing more autonomously. A picture of the setup taken during one of the measurement campaigns is shown in Fig. 1b. The main components

are described below:

- **Vehicle pass detection:** The exact passing time of the vehicles is of great significance for automated post-processing. This is especially the case if several vehicles pass by the measurement location and they have only a small spacing between them. The exact passing time is required during data post-processing to resolve the different plumes correctly. Important variables related to the vehicle condition during the passing are the speed and acceleration. These are required





to determine the vehicle specific power (VSP) (see Appendix A3). Emissions from passing vehicles strongly depend on the engine load conditions. Therefore, they must be treated accordingly (Bernard et al., 2018; Davison et al., 2020). For this purpose, we deployed custom-built light barriers in this study to measure the passing time, speed and acceleration of the passing vehicles. Using light barriers restricts the measurement location to single-lane roads or roads with islands between the lanes. Alternatively, vehicle detection can be performed with radar, video, or LiDAR systems.

– **License plate recognition:** Vehicle technical data are required for several post-processing steps which are described in more detail in the data analysis section later on. Automated number plate recognition (ANPR) systems are commonly used for license plate detection. Depending on the system, additional attributes such as the vehicle pass time or acceleration can be measured. Attention must be paid to the ANPR camera performance, as several influencing factors can exist. License plates are often dirty (especially in winter) or the ANPR camera may not able to correctly detect all the plates of 
the passing vehicles (especially if they pass within short intervals). This impedes data post-processing and underlines the importance of accurate vehicle pass detection. Based on our practical experience, we recommend that the vehicle pass time be detected separately from acquiring the license plate data.

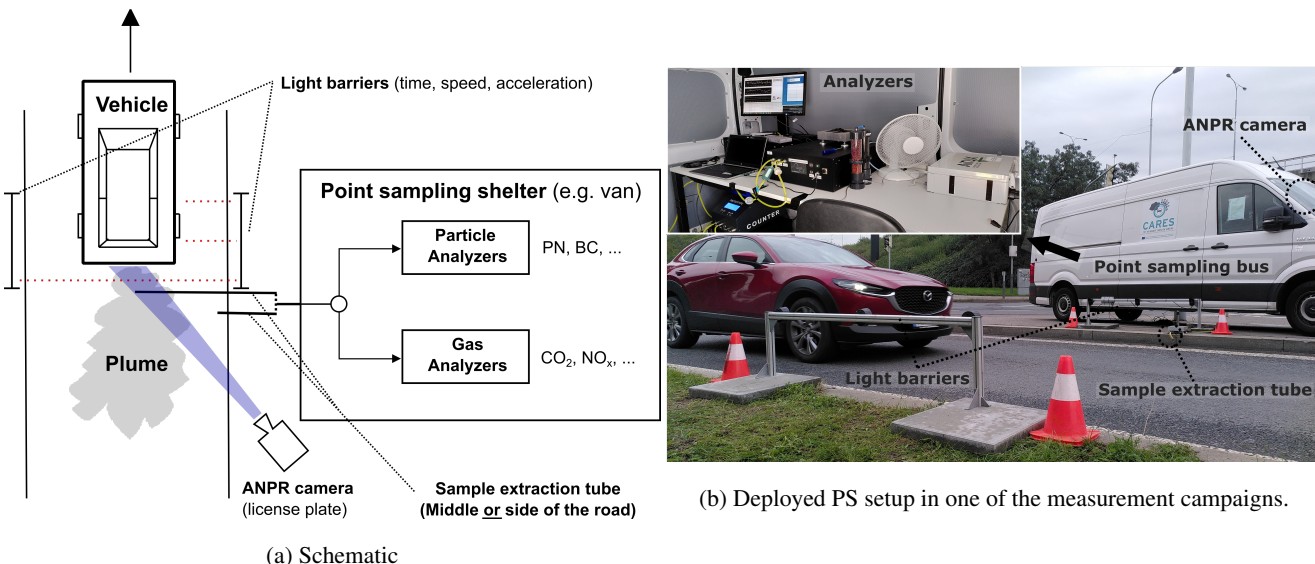

(a) Schematic

(b) Deployed PS setup in one of the measurement campaigns.

**Figure 1.** Schematic (**a**) and picture (**b**) of the proposed PS measurement setup, highlighting the required equipment.

– **Emission measurement:** The emission measurement can be split into two main parts: First, the emissions are sampled and second, these are subsequently analyzed with the employed instrumentation. A schematic of the emission measure-
ment setup used during one of the campaigns can be found in Appendix B.

     – Sampling: The importance of sample extraction is often underestimated. In PS, the sampling is usually performed with a simple tube which collects the diluted exhaust from the passing vehicles (Hak et al., 2009; Hallquist et al., 2013; Liu et al., 2019; Zhou et al., 2020). The position of the sample inlet can either be in the middle of the road by



fixing the tube directly on the road or on the side of the road. The position of the sampling inlet strongly influences the strength (dilution) of the measured plume and even determines whether the plume can be captured at all. In general, the closer the sample inlet is to the emission source (tailpipe) the smaller the dilution and the higher the capture rate are. We found typical dilution factors between 100 and 500, which is in good agreement with literature (Hak et al., 2009). In addition, the length of the sampling line must be considered in relation to the sample flow. The pressure drop should be minimal and the losses must be taken into account, especially when performing PM measurements (Kulkarni et al., 2011). Attention should also be paid to the material of the tubing. We use tygon tubing for particle measurement because of the flexibility and low particle losses (Giechaskiel et al., 2012).

– Instrumentation: When selecting analyzers for PS applications, attention must be paid to several aspects. Hak et al. (2009) mentioned in their PS experiments that the small dynamic range of the condensation particle counter (CPC) used constrained the PN measurements. Therefore, they used a dilution volume which extended the measured emission concentration peaks and the fall time of the signals to 5-15 s. The dilution and the relatively large response times of the particle instruments limited the operation to low-traffic situations. Based on the recommendations by Hak et al. (2009) and our own experiences, we have defined the most important requirements for the instruments, which are stated in Table 1. These must be respected to avoid significant problems in PS. In addition, recommendations are given for the different requirements. The emission events associated with the passing vehicles are of very transient nature. To capture these events, instruments must have a fast response time ($t_{90} < 1\text{-}2$ s) and a high time resolution (at least 1 s). In PS, the sampled emissions are highly diluted. Therefore, small concentrations must be resolved. To accurately measure the varying concentrations, the instruments must have a high dynamic range. The measured concentrations can be within a range of four orders of magnitude and depend on the vehicle type, engine state, sampling position and environmental conditions, as well as other factors. It is also important to ensure that the species of interest is measured with minimal cross-sensitivity to other compounds. Therefore, instruments with qualified measurement principles should be selected. Environmental conditions (e.g., temperature, relative humidity, background (BG) concentrations) differ depending on the measurement location, time and season of the year and care should be taken to ensure that they do not affect the instruments. RES campaigns often last for long periods of time (several weeks or months). Therefore, instruments must be stable over the long term. Due to restrictions in the use of calibration sources such as gas bottles or particle sources, the instruments should feature stable calibration over periods of weeks even under harsh environments. Instruments which do not require in-field calibration are preferred. To perform measurements under all conditions, an instrument housing is required, which can be a small shelter or a measurement van.

– **Monitoring of environmental conditions:** It is advantageous to make additional measurements of environmental conditions at the measurement location. Local monitoring of wind speed and direction provides information relevant for the sample extraction and can improve the post-processing. Taking measurements of precipitation, ambient temperature and relative humidity can also provide meaningful information and can help to understand abnormalities. We used data from



| Problem | Instrument requirement | Recommendation |
|---|---|---|
| Transient nature of emission events | Short ($t_{90}$) response time | $\leq$ 1-2 s |
| | High time resolution | $\leq$ 1 s |
| High exhaust dilution in ambient air | Low limit of detection (LoD) (at 1 s time resolution) | BC: 1 $\mu$g m$^{-3}$<br>PN: 1,000 # cm$^{-3}$<br>CO$_2$: 5 ppm<br>NO$_2$: 2 ppb<br>NO$_x$: 5 ppb |
| Varying concentrations | High dynamic range | BC: 0 - 2 mg m$^{-3}$<br>PN: 0 - 2e6 # cm$^{-3}$<br>CO$_2$: 0 - 3,000 ppm<br>NO$_2$: 0 - 2,000 ppb<br>NO$_x$: 0 - 5,000 ppb |
| Interfering species | Minimal cross sensitivity | Qualified measurement principle |
| | Minimal artefact formation | For PN: Solid particle number (SPN) measurement |

**Table 1.** Instrument requirements, problem statements and recommendations for PS emission measurements of selected particle metrics and gases.

weather stations either in the area or directly at the PS site. It is advantageous if the weather data is available directly from the measurement location.

## 2.2 Data analysis

The data analysis deals with the determination of representative EFs from the collected measurement data of the captured vehicles. The following aspects must be taken into account:

- Handling and harmonization of data (concentration time series) collected with various instruments.

- Consideration of measurement parameters such as sampling delay or instrument response times.

- Detection and separation of the plumes from the passing vehicles.

- Relation between vehicle pass (time), concentration time series and license plates.

- Dealing with changing environmental conditions (e.g., BG concentrations, other emission sources, weather conditions) that can affect the measurements.

In order to deal with the challenges listed above, a comprehensive data analysis framework was developed. The developed procedure has been divided into three major processing steps namely the pre-processing, the emission event processing and the emission analysis and statistics. The pre-processing reads the raw time series files from various instruments and the recorded



data from the light barriers (time, speed, acceleration) and prepares them for the next processing steps. These data are analysed in the emission event processing part of the procedure. The core of this procedure is our peak detection algorithm (TUG-PDA) which is applied to assign the captured emissions to the passing vehicles. The EFs are then calculated in the emission analysis step, and statistics are performed to subsequently evaluate the EFs. An overview of the data analysis procedure is visualized in Fig. 2. The software framework is designed for modularity and extensibility. New instruments and measurement campaigns can be easily integrated into the framework by copying existing instruments or campaigns and adjusting the parameters. The data analysis is implemented in Python by using common libraries such as Pandas, NumPy, Matplotlib, or SciPy.

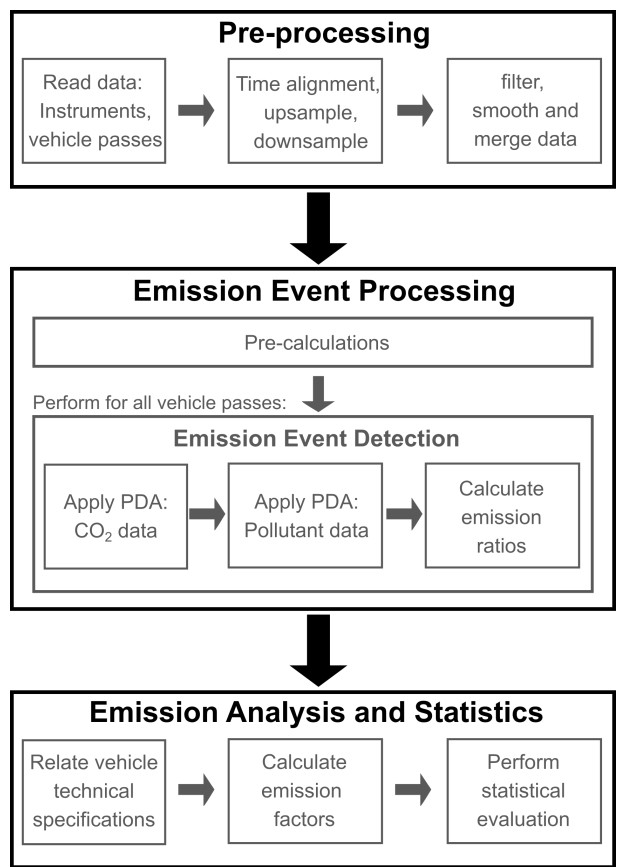

**Figure 2.** Overview of the PS data analysis procedure.

### 2.2.1 Pre-processing

In PS campaigns, various instruments are often used to measure different exhaust components (Hallquist et al., 2013; Wang et al., 2015; Ježek et al., 2015; Liu et al., 2019; Sugrue et al., 2020; Zhou et al., 2020). Because various instruments are used, different data formats need to be handled and the time resolution can vary between the devices. These heterogeneous datasets are harmonized into one composite dataset in order to appropriately process the data by using one data analysis procedure. After





reading the raw time series from the individual instruments, a time alignment of the data is performed. Different instruments
have varying response times which depend on the instrument response function, sample flow and sample tube connection to
the instruments. These differences must be compensated for. We determine the response times from short manual pollution
peaks (e.g., with a lighter) and align the instrument responses based on this event. The response time to the emissions of
individual vehicle passes varies depending on the sampling position, exhaust pipe position and environmental conditions. We
align the concentration time series data to the vehicle passes which cause the fastest response (e.g., from vehicle with tailpipe
on the same side as the sample extraction). Emissions e.g., from vehicles whose tailpipe is on the opposite side of the sampling
position are sampled with a delay of a few seconds. In addition, sampling delays between the sample inlet and the instruments
must be compensated for. To simplify data processing, the time resolution of the $CO_2$ data is equated to the time resolution of
the pollutant data (default time resolution of 0.5 s). Having higher time resolution makes it easier to resolve emission events
on smaller time scales. This enables vehicle passage data to be processed down to a spacing of only a few seconds, as long as
the instrument's response time permits. As final pre-processing step, error-prone data samples (e.g., outliers) are filtered out,
the instrument responses are adjusted to each other (by smoothing), and the datasets are merged.

### 2.2.2 Emission event processing

To determine the EFs, the transient emission events associated with the passing vehicles must be properly detected. For that
purpose, a dedicated algorithm was developed. The TUG-PDA separately processes the $CO_2$ time series and the time series of
the measured pollutants (e.g., BC, PN, $NO_x$) since PM and gaseous emissions can occur at a different time. The time series
data of $CO_2$ and pollutants do not have to be perfectly aligned to each other ($\pm$ 1.0 s, see Appendix E) and the response
functions of the used instrumentation do not have to be perfectly matched. At the same time, care must be taken to ensure that
the $CO_2$ plume detected of the passing vehicle is related to the pollutant emission detected. Therefore, checks are implemented
which compare the duration of the integrated $CO_2$ and pollutant data and verify if the areas overlap appropriately. The $CO_2$
data signals are thus processed prior to processing the pollutant emission data using the TUG-PDA. The TUG-PDA relies on
the time series data of the measured analyte (e.g., $CO_2$, BC), the passing times of the vehicles and the defined thresholds for
the emission detection as input. If pollutant emissions are processed, the event results from the corresponding $CO_2$ processing
(start and stop time of the integration) are also required.

Fig. 3 shows a flow chart of the TUG-PDA with the main processing steps of the algorithm. The TUG-PDA loops through all
the vehicle pass data. First, when a new vehicle pass is fetched, it is checked whether the distance ($\geq$ 3 s) to the next vehicle
pass is sufficient. If this is not the case, the processing for the current vehicle is stopped and the algorithm proceeds to the
next vehicle. At this small spacing, there is a large uncertainty that emissions will be attributed to the wrong vehicle due to
differences in the sampling delay between vehicles. The vehicle pass time is used as a starting point for the plume detection.
The TUG-PDA searches from this point onward for a sequence of positive concentration gradients above a defined threshold
of the processed analyte (visualized in Fig. 4). If a sequence of positive gradients has been found, the starting point of the
sequence is used as the starting point for the plume integration. Prior to the integration, the BG concentration is determined.
We observed that the minimum concentration directly before the vehicle pass fits best because it represents the actual condition.





Similar approaches were used in literature to determine the BG concentration (Ban-Weiss et al., 2008; Wang et al., 2015). One special case for BG determination is when vehicles pass the measurement point within a short period of time and the plumes overlap. If this is the case, an interim value is taken, i.e. a value between the previously described BG value and a concentration taken within the last minutes which is not influenced by vehicles. After determining the BG, the concentration of the exhaust plume is integrated until one of the defined stop conditions is reached (see Fig. 3):

- A maximum event duration (25 s) is exceeded.

- The concentration falls below the BG concentration.

- A subsequent vehicle pass occurs, and the concentration gradient increases.

After integration, several tests are performed to determine whether the captured emission event is valid:

- First, the duration of the integrated plume is reviewed to see if it meets a defined minimum value (in this study, three seconds). If this is the case, different tests are performed for either $CO_2$ or the examined pollutant:

  - In case of processing $CO_2$, the integrated area requires a minimum concentration (40 ppm s).

  - In the case of examining a pollutant, tests are performed to verify whether the time frames of the integrated areas for the examined pollutant and $CO_2$ coincide. It is not allowed that the duration of the pollutant area exceeds the area of $CO_2$.

If the conditions are met, the emission event is considered to be valid, and the integrated concentration and the time information (start, stop) for the emission event is stored (highlighted in green in Fig. 3). The TUG-PDA continues processing the next vehicle pass. When processing the pollutant data, a special case is implemented in order to consider low emitters (vehicles with small pollutant emissions). If a substantial concentration gradient is not found, and the minimum event duration criteria is not fulfilled, the pollutant concentration is integrated over the duration of the captured $CO_2$ event associated with the passing vehicle (highlighted in blue in Fig. 3). After the TUG-PDA has finished the processing, the emission ratios (ER) (see Sect. A1) are calculated. The TUG-PDA is fully configurable with various adjustable parameters such as start time range, thresholds, or minimum number of required data samples.

An example of the data processing of the TUG-PDA for $CO_2$ and BC emission concentrations recorded of two passing vehicles is shown in Fig. 4. The passing time is indicated by yellow, vertical, dashed lines. The TUG-PDA algorithm starts searching one second before the vehicle pass for a significant rise in emission concentration. The emission concentrations which are assigned to the first shown vehicle pass are highlighted by the integrated areas of $CO_2$ (shaded in blue) and BC (shaded in black). In this case, the TUG-PDA stopped the integration, because another vehicle pass was identified, and an increasing gradient was found. The plumes from the passing vehicles can be easily separated and an ER can be calculated in this case. The BG concentrations of $CO_2$ and BC are subtracted from the integrated areas. More information on the capabilities and limits of emission separation of densely driving vehicles is provided in Appendix C.



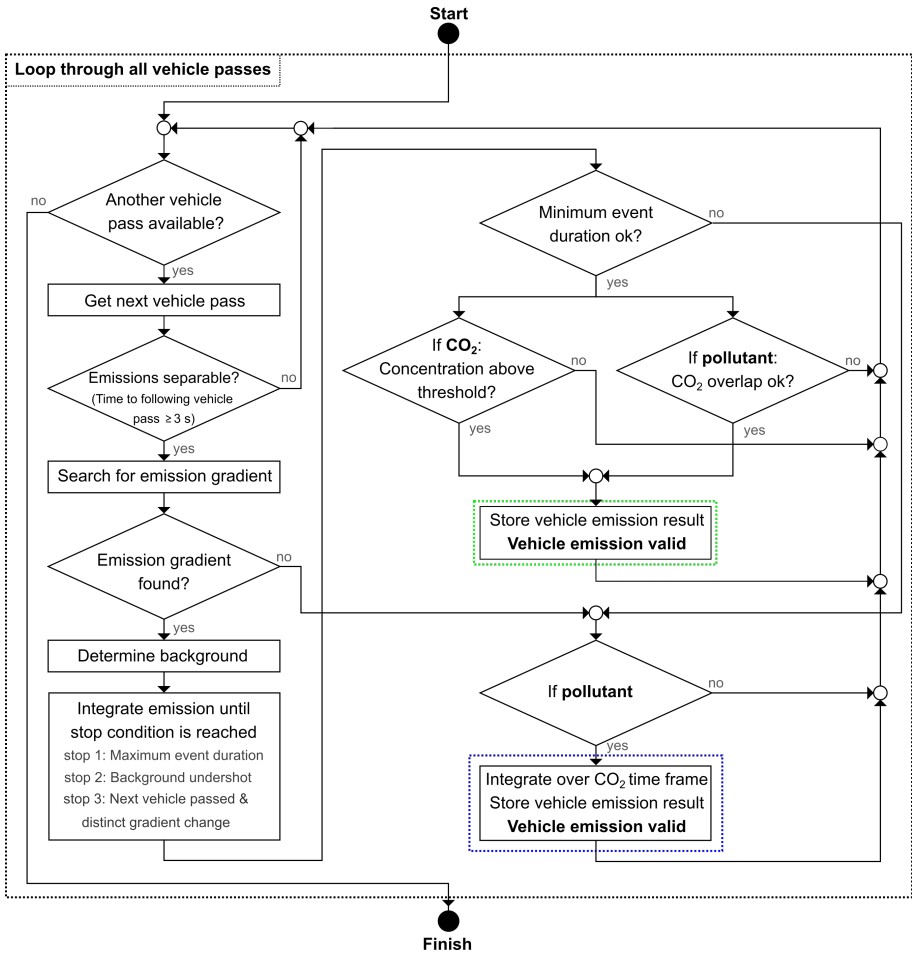

**Figure 3.** Emission event detection - Flow chart of the TUG-PDA. $CO_2$ and pollutant (e.g., BC, PN, $NO_x$) emissions are processed individually. Highlighted in green: outcome for pronounced plumes. Highlighted in blue: outcome for low emitters.

Alternative methodologies exist for emission processing in PS. The captured $CO_2$ and pollutant emissions are commonly integrated over the same time frame (Ban-Weiss et al., 2009; Ježek et al., 2015; Liu et al., 2019; Zhou et al., 2020). Automated PS emission processing algorithms can also be found in previous studies. Wang et al. (2015) presented a plume identification algorithm that takes different approaches in the case of plume separation (minimum plume length of 10 s) or low emitter detection. Another new approach proposed by Farren et al. (2023) is the so-called rolling regression method. This algorithm simplifies data processing by calculating the ERs for three consecutive data samples, which makes the BG determination redundant. This is a particularly promising approach for short emission events. One challenging aspect is that the instrument responses for $CO_2$ and measured pollutants must be perfectly matched when taking this approach. The applicability of this approach to evaluate PM pollutants still needs to be studied.





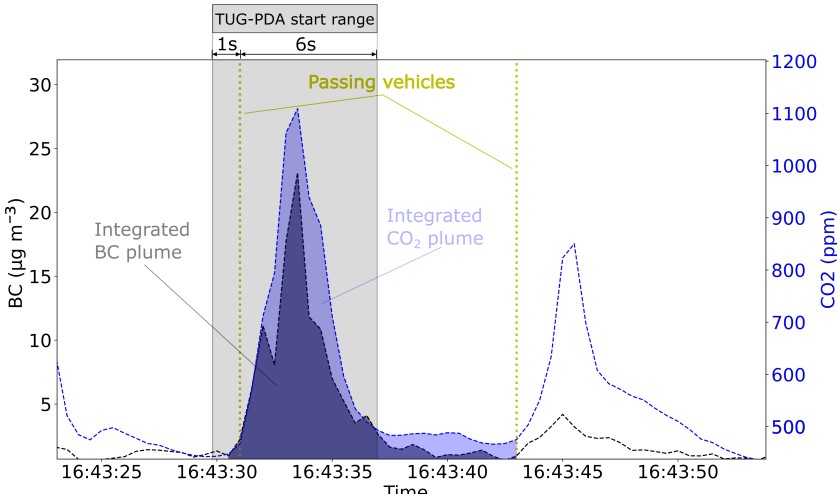

**Figure 4.** Time series example for the sampled PS data (BC, $CO_2$) from two vehicle passages. The integrated areas of the $CO_2$ and BC emission concentrations are highlighted for the first passing vehicle using the TUG-PDA as described in Sect. 2.2.2. The start time range of the algorithm is indicated for the first vehicle pass. The default TUG-PDA start range is -1 s to 6 s after vehicle pass.

### 2.2.3 Emission analysis and statistics

Once the ERs of passing vehicles have been determined, further analysis requires the vehicle technical data. This is usually
obtained from government organizations. It is important to respect the privacy of the license plates captured, which varies from country to country. Several details from the vehicle technical data are required during the emission analysis to calculate EFs and to perform further statistical analysis. The fuel type (e.g., gasoline, diesel) must be known to calculate fuel-based EFs. For the calculation of distance-based EFs, the $CO_2$ emissions as measured during the type approval process for the vehicle model are required. In most European countries, the European emission standard class is used to classify vehicles according to their
emission limits. Information such as the manufacturer or vehicle category are used to perform detailed evaluations of fleet emissions and to identify unusual emission patterns, such as individual high emitting vehicle models or manufacturers.

As part of our data post-processing procedure, the vehicle technical data as requested from the authorities and detected by the ANPR camera must be related to the ERs. These are then assigned to the passing time as gathered with the light barriers. We use the speed and acceleration information of the passing vehicles to match the passing time with the detected license plate.
This generally sounds like a simple task. However, not all license plates are correctly detected by the ANPR camera for various reasons (e.g., dirt, poor light conditions, too little distance between the vehicles). This makes the task of correctly matching the data from the ANPR camera and light barriers challenging, and specially for vehicles that follow each other closely. In our setup, the ANPR camera is mounted in the front cabin of the measurement van (see Fig. 1b), allowing the license plates to be detected about 2-3 s after the vehicle passes the light barriers.



# 3 Results and discussion

## 3.1 Capture rate

In RES, the proportion of valid measurement records is a significant indicator. We call this indicator the capture rate (CR), which is the ratio between the number of vehicle passes for which valid EFs can be calculated and all vehicle passages:

$$CR = \frac{\#\,valid\,EF}{\#\,all\,vehicle\,passes}. \tag{1}$$

What is considered as a valid measurement is always subjective. We consider the calculated emissions to be valid if the plume from the passing vehicle was properly captured and an EF can be calculated. This is the case if the following conditions are considered to be true:

– The integrated $CO_2$ plume is greater than a specified threshold. In this study, 40 ppm s was used.

– The emissions of the passing vehicle can be separated from those of other vehicles. This is not the case if the plumes cannot be separated or if the emissions can't be unambiguously assigned to one vehicle.

## 3.2 Evaluation of influencing factors

In the following part, the most important factors influencing PS measurements are discussed and the resulting impacts are shown on basis of around 100,000 vehicle emission records gathered during 4 measurement campaigns. The measurement campaigns were conducted in the Netherlands, Italy, Poland and Czechia. The results include data from 9 measurement locations. In several figures (e.g., Fig. 6, 7, 8a, 8b), the results from the different measurement locations are labeled with numbers. Different sampling positions or traffic situations were evaluated on individual locations. These are labeled with x.x (e.g., 1.1). This should facilitate the interpretation of the results and the comparison between the different impact factors, as they were not determined independently. For the evaluation of the influencing factors, mainly data from the newly developed black carbon tracker (BCT) were used. The BCT measures BC with a photoacoustic based sensor cell and $CO_2$ with a non-dispersive infrared (NDIR) sensor integrated into one device. The device was developed based on the recommendations listed in Table 1 as part of the CARES project (Knoll et al., 2021). The impact of misaligned measurement data on the resulting EFs is discussed in Appendix E.

### 3.2.1 Sampling position

The approach used to collect the exhaust has a major impact on quality and strength of the signal. In PS, the sample is commonly extracted either from the side or the middle of the road. The integrated $CO_2$ concentration of the captured plume of the passing vehicles serves as a marker for comparing sampling positions. Therefore, we compared the $CO_2$ concentrations





when we measured either from the middle, the left, or the right side of the road. If the sample was taken in the middle of the

road the sampling tube was fixed directly to the roadway. All three sampling positions were used at least three times during the measurement campaigns. Distributions of mean $CO_2$ concentrations of the three sampling positions were calculated using the Monte Carlo method by drawing 500 samples of the measured $CO_2$ concentrations of passing vehicles from each measurement position 1,000 times. Sampling from the middle of the road gives on average clearly higher signals as compared to sampling from either side of the road, with a mean and $\sigma$ of 781 ppm and 1867 ppm, respectively. Sampling from the left (mean: 599 ppm,

$\sigma$: 914 ppm) delivers on average a higher signal as compared to sampling from the right side (mean: 554 ppm, $\sigma$: 995 ppm). A Gaussian distribution was assumed and fitted to the three datasets (Fig. 5a). In general, the closer the sample is extracted to the exhaust source, the stronger the captured signal is. In most regions in Europe, sampling from the left is favored over the right side. Vehicles from manufacturers in Europe (e.g., VW, BMW, Mercedes, Fiat) have usually the tailpipe on the left-hand side, unlike manufacturers in Asia or the United States (e.g., Toyota, Kia). We also evaluated the mean CR of the $CO_2$ plumes from

the three sampling positions (Fig. 5b). A direct relation is observed between the $CO_2$ signal strength and the CR. A higher $CO_2$ signal generally leads to a higher CR. The highest CR can be achieved if the sample extraction is performed from the middle of the road. By using this central setup, a $CO_2$ plume could be captured for an average of 41.3 % of the vehicles. This relates well to Hak et al. (2009), who reported a CR of about 50 % with their setup sampling from the middle of the road. Sampling from the left delivers on average a CR of 31.6 % as compared to 23.2 % if the sample is extracted from the right

side. The variance for measurements in the middle of the road is smaller than at the roadside. At two measurement locations (3 and 7), the sampling was conducted from both, the roadside and from the center of the road. Two to three times higher CRs were obtained when sampling from the middle of the road (Fig. 6 and Fig. 7). Sampling from the center of the road is less influenced by location and vehicle characteristics such as road width and tail pipe position. The influence of wind conditions is less because the exhaust is sampled in close proximity to the source. In addition to the influence of the sampling side, the

sampling height also has a major impact on the sample extraction. Higher CRs and stronger $CO_2$ signals are achieved at lower sampling inlet heights. This is particularly evident at measurement locations 1 and 2, where the position of the sample inlet was shifted by 3-4 m with slight differences in sampling height (Fig. 6) and road width (Fig. 8b).

### 3.2.2 Measurement location

Special care must be taken when selecting suitable measurement locations. The selection of the measurement location influ-

ences the following aspects:

- Road properties (single or multi-lane, lane width, gradient)

- Traffic conditions (traffic flow, distance between passing vehicles, number and type of vehicles)

- Vehicle operating conditions (VSP)

- Influence of environmental conditions

- Cross-interference from other pollution sources





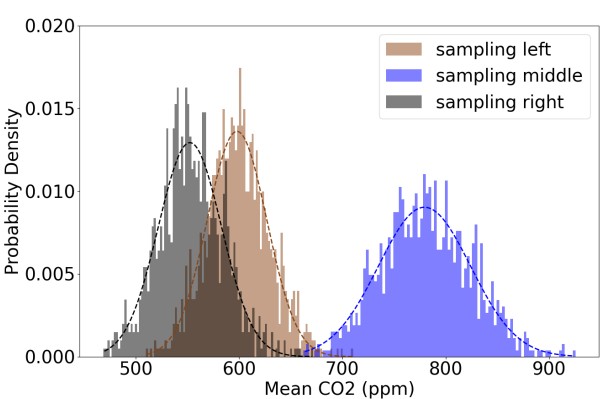

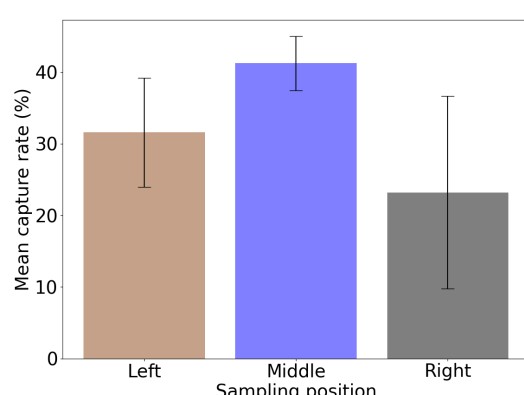

(a) Distribution of mean $CO_2$ concentrations of the three sampling positions.

(b) Mean capture rate at the three sampling positions including standard deviation.

**Figure 5.** Mean $CO_2$ concentrations (BG concentration already subtracted) and mean capture rate for the three evaluated sampling positions (left, middle, right).

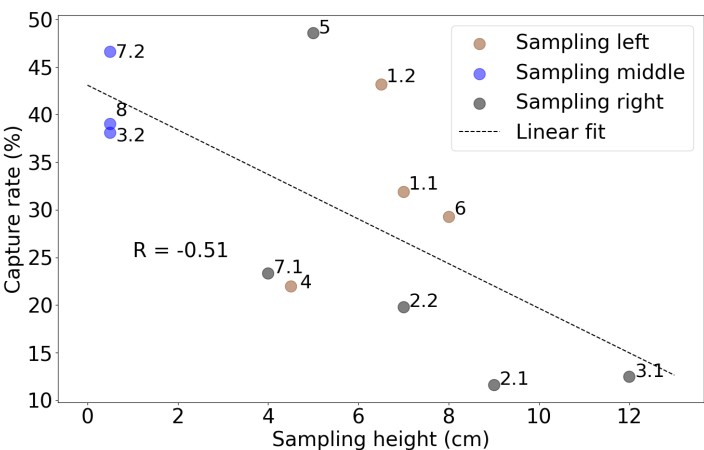

**Figure 6.** Capture rate as a function of the height of the sampling inlet. The numbers represent the different measurement locations. The sampling position (left, middle, right) is highlighted. At measurement sites 1 and 2, sampling was conducted from two positions shifted by 3-4 m on the same side with slight differences in sampling height and road width.

One selection criteria of PS campaigns is often the number of vehicles per site and day. Conducting campaigns on highly frequented roads guarantees a high number of vehicle passes. This is to a certain extent beneficial, as it allows for the collection of a large number of emission records. If the traffic density is too high for PS, the emissions from the individual passing vehicles cannot be properly resolved because they superimpose (e.g. Fig. C1b). Not only the traffic density, but also the general traffic



flow must be considered. Measurements are often performed after a crossroad or traffic light. Such conditions can lead to a high number of vehicles passing within a short period of time and a short distance of each other. This prevents emissions from being properly resolved. Therefore, we evaluated the CR as a function of the median vehicle distance at different measurement locations. The CR generally increases with median vehicle distances at the measurement locations (Fig. 7). A smaller distance between the vehicles makes it more difficult to separate the emissions. It is noticeable that even in relatively dense traffic

(median vehicle distances 3.3 - 6.2 s), a high CR (38 - 47 %) can be achieved if the sampling is done from the middle of the road. The time interval between the vehicle pass and the corresponding exhaust plume reaching the sample inlet is smaller if the sample is extracted from the road center. At measurement location 6, we evaluated the CR for workdays (Fig. 7, 6.1) and weekends (Fig. 7, 6.2). During weekdays, the CR was rather low (26 %) with about 6.500 vehicles per day and a median vehicle distance of 2.5 s. On weekends, the traffic density was much lower with about 3000 vehicles per day and a median vehicle

distance of 5.6 s, so that a significantly higher CR (35 %) could be achieved. Appendix C shows two dense traffic situations in which emissions for all passing vehicles can either be separated or not separated using the TUG-PDA. Higher CRs can be achieved when tests are performed for test vehicles or in isolated environments such as test tracks. Ježek et al. (2015) were able to capture 125 out of 150 plumes during test track experiments when sampling from the roadside. Wang et al. (2015) reported a CR of 70 % for roadside sampling and 46 % for sampling 15 m from the road for nighttime measurements with a test vehicle.

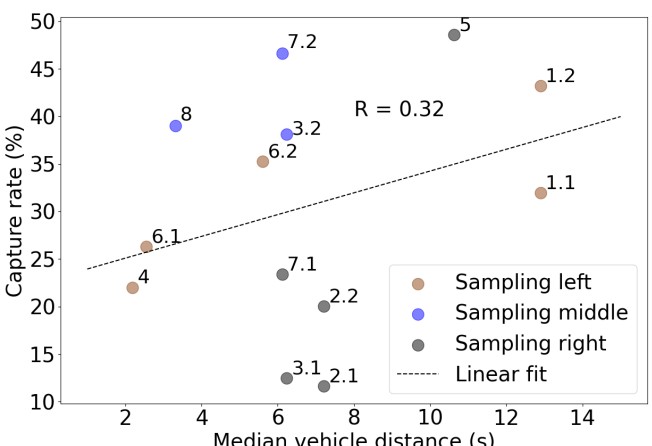

**Figure 7.** Capture rate as a function of the median vehicle distance at different measurement locations. The measurement locations are labeled by numbers. At measurement locations 3 and 7, the sampling was conducted from both the right side (3.1, 7.1) as well as from the center of the road (3.2, 7.1).

In order to select the measurement site, the road itself and the topography must be evaluated. We examined the influence of the VSP on the CR for the three sampling positions (left, middle, right). For this investigation, only speed, acceleration and the road gradient were used to calculate the VSP (see Appendix A3). The determined VSP values for the different measurement locations were clustered and averaged. We observed a small impact of the VSP on the CR (Fig. 8a). The CR increases slightly





with increasing VSP regardless of the sampling position. A certain engine load (e.g., VSP > -5 kW t$^{-1}$ according to Bernard
et al. (2018)) is required for the measured vehicles, which can be accomplished in locations with a positive road gradient or at
locations where vehicles accelerate (e.g. road crossings, slip roads). Measurements are often made after road crossings, where
passing vehicles can be assumed to accelerate. This aspect should be critically assessed, since traffic conditions must be taken
into account, as previously described. Roads with declining gradients should generally not be chosen due to lack of engine
load. The road type must be considered in terms of space for the measurement setup and cross interference from other vehicles.
In general, single-lane roads are preferred, as well as two-lane roads where the measured direction has a positive gradient.
Vehicles driving on the opposite lane have a negative VSP and thus a low engine load. Along with the VSP, the width of the
road has a non-negligible impact on the sampling. When sampling from the side, the distance from passing vehicles strongly
impacts the sample extraction due to increased dilution and greater dependability of tail pipe position or wind conditions. The
CR as a function of the lane width is depicted in Fig. 8b for measurement locations where the sample extraction was performed
from the side of the road. At two locations (1, 2), the sampling was conducted at two positions (1.1, 1.2 and 2.1, 2.2) at the
same roadside with differences in road width and sampling height. For both, it can be seen that a smaller road width and a
lower sampling height (see Fig. 6) lead to a higher CR. Measurement locations where the sampling was done from the right
side (2, 3, 5, 7) generally have a rather low CR. An exception is location 5, where the highest CR of all measurement sites was
achieved. Location 5 stands out with good characteristics of all influencing factors such as a small lane width (3.5 m), a low
sampling height (5 cm) and a high median vehicle distance of 10.6 s (∼ 2.500 vehicle per day). This highlights the importance
of selecting appropriate measurement sites.

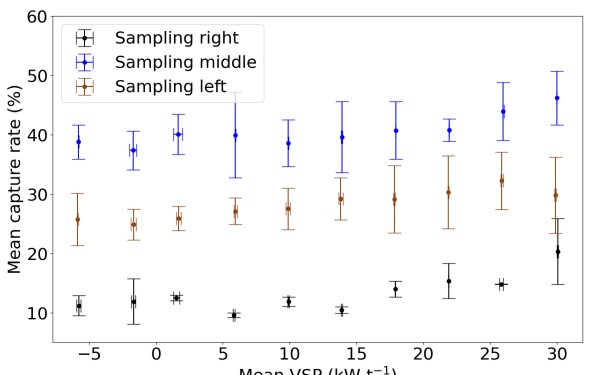

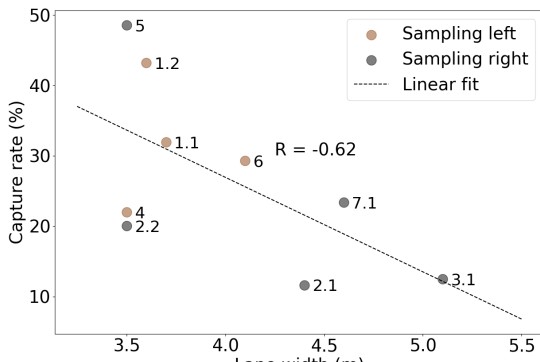

(a) Mean capture rate as a function of the VSP from different
sampling positions.

(b) Capture rate as a function of the lane width at different mea-
surement locations.

**Figure 8.** Evaluation of two impact factors of the measurement location and their influence on the capture rate.

Different measurement locations may be accompanied with varying environmental conditions such as wind or BG concen-
trations of the measured species. In Fig. 9, 30-minute averaged BG concentrations for BC and CO$_2$ are shown for three different
measurement sites over a time period of 24 hours. At Location 1, rather stable BC concentrations can be noted along with a



distinct increase in the $CO_2$ concentration during the morning traffic period. In contrast, very high BC BG concentrations were measured at Location 4 on the presented measurement day accompanied by varying BG $CO_2$ values. Such different conditions depend on the season, meteorology, traffic density and other emission sources like industry and must be taken into account. High or varying BG concentrations can impact instrument performance. The BG concentration must be compensated for in the emission calculation and results can vary widely depending on the BG concentration used. In addition to using the con-

centration directly before the emission peak, two other approaches to determine BG values were evaluated. 1) A statistically determined BG values was calculated by removing emissions greater than the 75th percentile of the used dataset. A moving average filter was applied to the resulting dataset and the minimum valued was used as BG concentration. 2) Usage of a median or average BG concentration during the last time window when no vehicle was passing. The two approaches proved to be less accurate considering all measurement conditions.

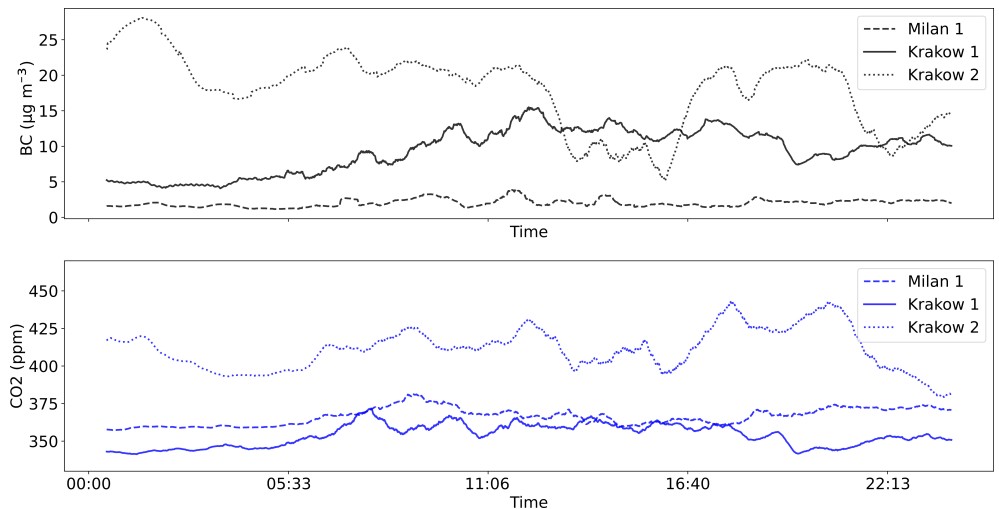

**Figure 9.** Background concentrations of BC (upper plot) and $CO_2$ (lower plot) from three different measurement sites. Concentrations are averaged with a half-hour running mean filter.

**3.2.3 Weather conditions**

Harsh weather conditions can have a substantial impact on RES measurements. For both PS and RES, the literature lacks detailed assessments that examine the effects of environmental conditions. Of particular interest are the dependencies related to precipitation and wind conditions. Commercial open-path RES systems have difficulties to measure during precipitation. During the measurement campaigns, a weather station was either located directly next to the PS site or in the vicinity. The

weather data used were available on at least hourly basis. To allow an unbiased comparison, only datasets were used where the compared meteorological conditions were present during the measurement campaigns.

In Fig. 10a, the CRs are compared during rainy and dry conditions. On average, a slightly higher CR was determined in 36.7 % of the cases under dry conditions as compared to 36.2 % of the cases during rainy weather. Similar values were also determined





for the average $CO_2$ plume of the passing vehicles. During dry periods, an average $CO_2$ plume of 630 ppm was measured as
compared to 631 ppm under wet conditions. We were particularly interested in discovering whether these conditions impacted
PM emissions. For this purpose, we compared the determined ERs for the passing vehicles regarding BC. Statistically, no
significant difference was observed between the EFs calculated under dry or rainy conditions with median values of 110 and
121 mg kg$^{-1}$ $CO_2$, respectively (Fig. 10b). The slight differences may result, for example, from different driving behavior in
wet conditions.

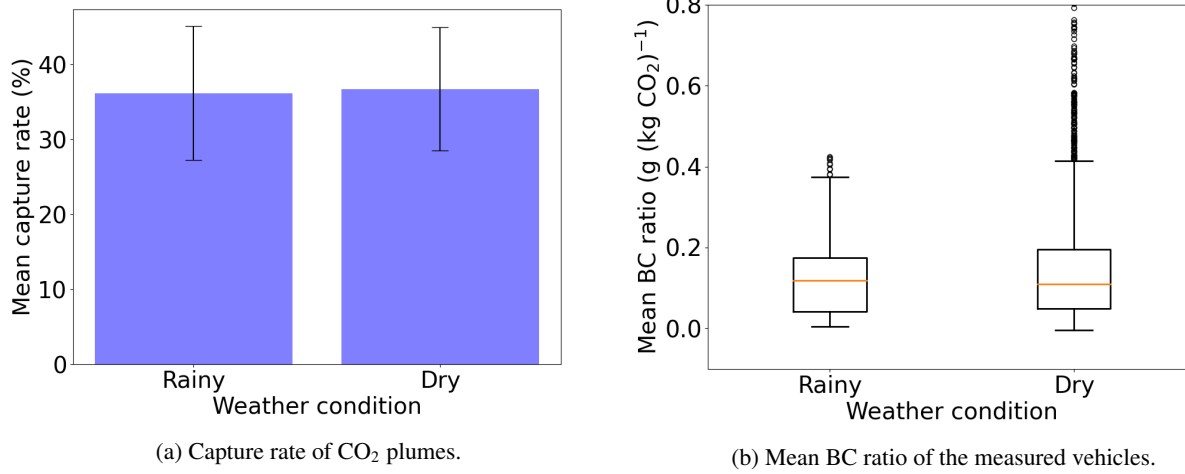

(a) Capture rate of $CO_2$ plumes.                    (b) Mean BC ratio of the measured vehicles.

**Figure 10.** Effect of precipitation on PS measurements. Measurements are compared for rainy ($> 0.05$ mm h$^{-1}$) and dry weather conditions.

Wind direction and wind speed affect the dilution and transport of the plume. The wind speed was segmented according to the
Beaufort scale (Singleton, 2008). Under rather calm wind conditions (0-11 and 12-19 km h$^{-1}$, BFT 1,2 and 3), no significant
impact could be observed. Under *moderate breeze* wind conditions (Beaufort scale, 20-28 km h$^{-1}$), a decrease in the CR is
noticeable. This trend continues under *fresh breeze* conditions (29-38 km h$^{-1}$), although only measurements from one PS site
for such conditions were available (Fig. 11a). This trend shows that the CR generally decreases with increasing wind speed. A
similar trend can be observed for the measured average $CO_2$ plume of the passing vehicles. A higher $CO_2$ signal (684 ppm) was
measured under calm conditions ($< 20$ km h$^{-1}$) than under windier (21-39 km h$^{-1}$) conditions (542 ppm). A similar influence
of wind speed on the CR was reported by Dallmann et al. (2011) in their top-down PS study for HDVs. They reported lower
CRs in June (61 % unsuccessful plume captures) than in November (36 % unsuccessful plume captures), where average wind
speeds were twice as high. In contrast to our results, they found that the dilution of the captured plumes was similar for both
wind conditions.
Not only the wind speed is relevant, but also the direction in which the wind blows the exhaust plume. We evaluated the impact
of the wind direction on the PS measurements under calm ($< 11$ km h$^{-1}$) and breezier ($> 11$ km h$^{-1}$) circumstances. For this





purpose, we separated the wind directions into wind blowing the exhaust plume towards the measurement location and wind blowing it away from the sampling point. The wind directions are indicated in Fig. D1 in the Appendix. A significant influence was observed at a rural measurement location (Fig. 11b). The CR is higher under calm conditions and when winds are blowing towards the sampling position. We performed the same evaluation in urban environments. Here, we could not observe such a trend with similar CRs regardless of the wind direction (Fig. D2). We assume that this is mainly related to differences between the local wind conditions (local turbulences) directly at the PS spot and the wind measured at the weather station. Generally, wind conditions in street canyons are much calmer than those in open spaces, which is beneficial for PS applications.

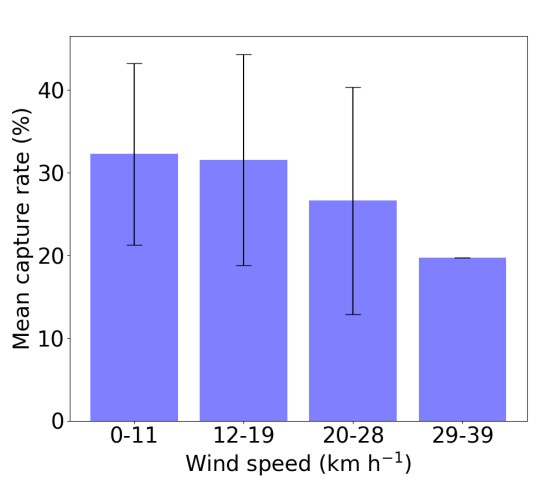

(a) Wind speed at urban measurement locations.

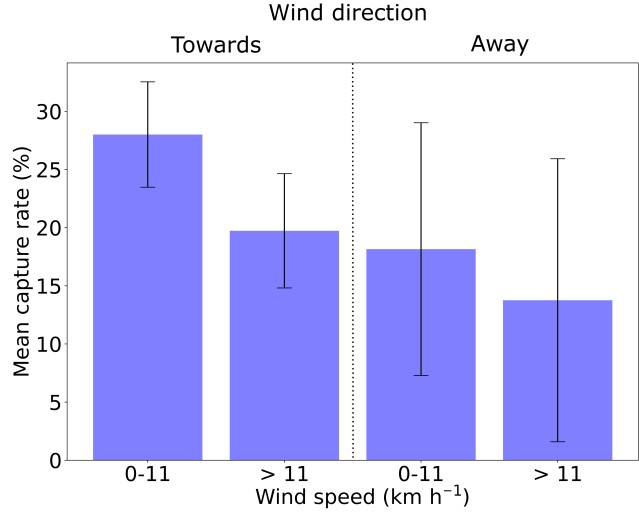

(b) Wind speed and direction at a rural measurement site.

**Figure 11.** Influence of different wind conditions on the capture rate.

The influence of temperature is investigated in Fig. 12 for low ($\leq$ 10 °C) and high temperatures (> 10 °C). Ambient temperatures ranged from -7.3 °C to 28.2 °C during the different measurement campaigns. No significant difference was observed with an average CR of 35.5 % at low temperatures and of 35.9 % at high temperatures. The effects of ambient temperature and humidity are not expected to have an impact on the PS measurement itself, if the instrumentation used are either properly stored or can perform measurements under such conditions. Ambient temperature is expected to have an impact mainly on the passing vehicles and their exhaust after-treatment systems (Kwon et al., 2017; Ko et al., 2019).

### 3.2.4 Instrument characteristics

Instrument characteristics (see Table 1) have a great influence on the quality of the measured emission data. Sugrue et al. (2020) compared high- and low cost BC and $CO_2$ sensors for their application in PS. They found that low-cost $CO_2$ sensors may be an adequate substitute for research-grade analyzers in contrast to low-cost BC instruments. In their conclusion, they also emphasized that sensors should be tested under field-conditions. As an example, we compared characteristics of the custom-



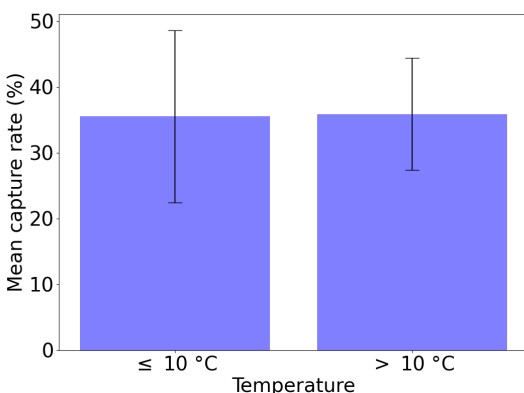

**Figure 12.** Influence of ambient temperature on the capture rate.

designed BCT with those of a commercially available Aethalometer AE33 (Magee Scientific). The Aethalometer AE33 is widely used in environmental science for BC measurements and source appointment and is commonly used in PS studies to quantify BC emissions (Ježek et al., 2015; Preble et al., 2018; Zhou et al., 2020; Sugrue et al., 2020). Laboratory measurements

showed a very good correlation ($R^2$ = 0.99) between the Aethalometer and the BCT. Comparable LoD (3 $\sigma$) were determined for both instruments, with values of 1.01 $\mu$g m$^{-3}$ for the Aethalometer and 1.12 $\mu$g m$^{-3}$ for the BCT. The LoD of the instruments defines the extent to which emissions can be resolved. This is of particular importance in order to accurately quantify emissions from vehicles that meet the latest emission standards. The $t_{90}$ response times of the two instruments were measured in the laboratory: 0.9 s for the BCT and 7 s for the Aethalometer. A small response time enables the separation of highly transient

emission events. This determines how close vehicles can drive to each other in order to be able to resolve the emissions. Fig. 13 shows two emission time series of the two instruments during one of the measurement campaigns. Two vehicles pass by the PS spot during the shown time frame with an interval of six seconds. The BCT responds quickly to the captured BC emissions from the first passing vehicle (V1). A distinct peak is noted where the measured concentration is again below 10 % of the peak concentration of the first vehicle when the second vehicle (V2) passed by. The emissions captured for the two vehicles

overlap, but they can be separated. In contrast, the Aethalometer response time is much slower and the maximum concentration is reached after the second vehicle (V2) has passed by. The emissions from the two vehicles cannot be separated in this case. This is an example that shows how important it is to select instruments with suitable characteristics for PS applications in dense traffic. Individual characteristics that do not meet the requirements of the application can severely affect the measurement data.

**3.3   Application example**

In total, for the city measurement campaigns in Italy, Poland and Czechia, it was possible to collect technical data from authorities for 66,803 of the recorded vehicles. The technical data sets collected were pseudo-anonymised to comply with the



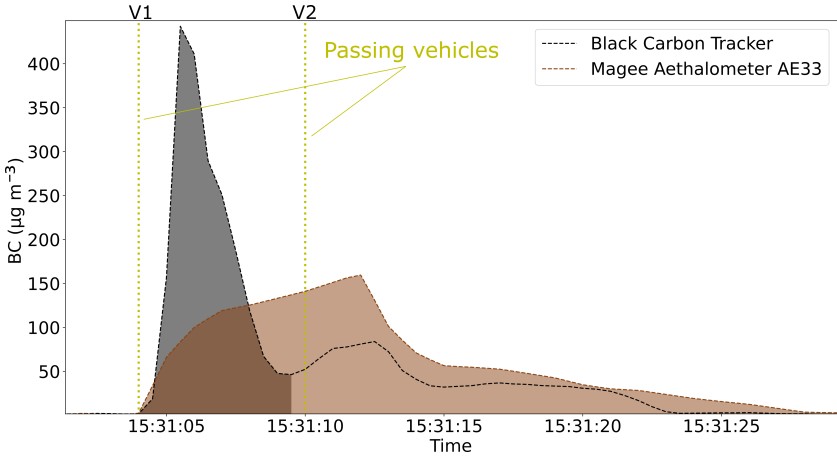

**Figure 13.** Emission concentration time series example of two instruments with different response times. Gray (Black Carbon Tracker) and brown (Magee Aethalometer AE33) shaded areas show the integrated areas of the BC emissions from the first passing vehicle as determined with the TUG-PDA.

data protection regulations of the individual countries. Based on the collected technical data sets, we determined with our data analysis framework (see Sect. 2.2) the emissions of 27,775[1] vehicles. Measurements were conducted with our described setup

(see Sect. 2.1). Several instruments were used in the campaigns to measure BC, PN and $NO_x$ EFs. The newly developed BCT was used to measure BC and $CO_2$, a custom designed diffusion charger (Schriefl et al., 2020) measured PN concentrations and an ICAD (Airyx GmbH, Horbanski et al. (2019)) was deployed for $NO_x$ and $CO_2$ measurements. A schematic of the emission measurement setup can be found in Appendix B.

### 3.3.1  Fleet composition and capture rate

The measurements were carried out in city centers, which is also reflected in the vehicle fleet. The vehicle types were classified according to the vehicle categories of the United Nations Economic Commission for Europe (UNECE). The largest share of vehicles measured were passenger cars (83.8 %). A much smaller share of L-type vehicles (1.6 %) and HDV and buses (0.8 %) were recorded (Fig. 14a, upper plot). We determined the CRs for the different vehicle categories to verify the ability to measure different vehicle types. The highest CR could be achieved for HDVs and buses (55 %), followed by passenger cars (35 %). The

CR of L-type vehicles, including motorcycles and scooters, was significantly lower with 27 % (Fig. 14a, lower plots). Previous PS studies (Dallmann et al., 2011, 2012) reported CRs for HDVs using top-down measurements from a bridge and a tunnel. In these studies CRs ranged from 12 % to 59 % for individual trucks and from 16 % to 44 % for groups of trucks. In general, it can be said that the CR depends on the exhaust flow rate of the vehicles. HDVs and buses have much greater exhaust flow rates than passenger cars or L-type vehicles. This is also reflected when looking at the average integrated exhaust plume of the different

vehicle categories. The average integrated exhaust plume of HDVs and buses (926 ppm s $CO_2$) was significantly higher than

---

[1]23,430 excluding multiple passes of the same vehicles





those of passenger cars (519 ppm s $CO_2$) and L-type vehicles (327 ppm s $CO_2$). A lower percentage of L-type vehicles is measured not only because of the smaller exhaust flow rate, but also because of the direction of the exhaust pipe. In contrast to HDVs, buses and passenger cars, the exhaust pipe for L-type vehicles often points upwards, which is disadvantageous when sampling from low heights. Looking at the distribution of the fuel type of the measured vehicles, a similar number of diesel

(45.5 %) and petrol (45.8 %) vehicles were measured. A small share of CNG, LPG or bi-fuel (petrol/diesel + CNG/LPG, 2.5 %) was captured (Fig.14b, upper plot). In contrast to the vehicle type, the CR is rather independent of the fuel type (Fig. 14b, lower plot). EFs could be determined for 35 % of diesel vehicles, which represents a slightly higher CR compared to 34 % of the petrol vehicles measured. This is mainly due to the fact that vehicles with a larger engine displacement (e.g. trucks or buses) are mostly powered by diesel engines, while smaller vehicles are mostly equipped with petrol engines (e.g. L-type vehicles).

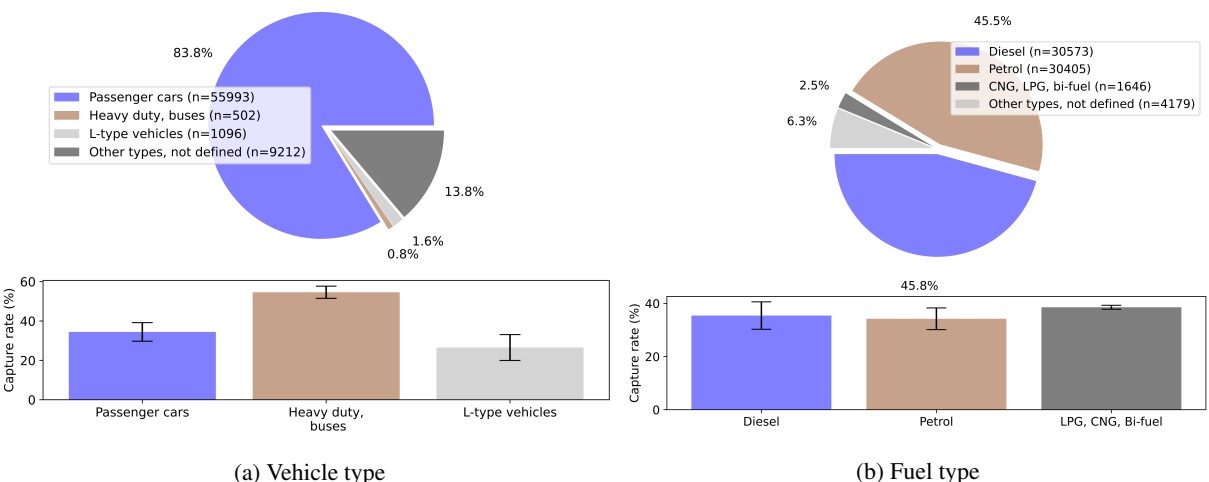

**Figure 14.** Measured vehicles split into vehicle categories and fuel type (upper plots). Capture rates for the different types are shown in the lower plots.

### 3.3.2 Fleet emission characteristics


Fuel-based EFs (see Appendix A1) were determined for the measured vehicles using the collected technical vehicle data. The fuel-based EF accounts for the larger total emissions from large vehicles such as HDVs and buses and makes all derived EFs comparable. Statistical evaluations were carried out for various vehicle categories and Euro emission standards (Fig.15 to Fig.17). Upper and lower whiskers represent the 97.5 and 2.5 percentile, respectively. The number of vehicles in each category

is indicated by the numbers in the brackets. Emissions from hybrid electric vehicles are included in the statistics. There are mainly two reasons for negative EFs. First, negative EFs result from low emitting vehicles where the determined background is higher than the measured emissions during the captured $CO_2$ plume. Second, the emissions of previously passing vehicles interfere with the measurement of the current vehicle.

BC emissions from petrol passenger cars (M1 category) decrease slightly from Euro 2 to Euro 6 emission standards, with mean




values between 87 and 150 mg kg$^{-1}$ fuel and median values ranging from 10.1 to 16.5 mg (kg fuel)$^{-1}$. For diesel passenger cars, BC emissions decrease significantly with increasing Euro emission standards and decreasing vehicle age from 1.82 g (kg fuel)$^{-1}$ (median: 0.87 g (kg fuel)$^{-1}$) for Euro 2 down to 0.078 g (kg fuel)$^{-1}$ (median: 7 mg (kg fuel)$^{-1}$) for Euro 6. The impact of the introduction of DPFs sticks out from Euro 5 onwards. Median BC emissions drop by a factor of 15 from Euro 4 to Euro 5 vehicles, from 0.19 to 0.013 g (kg fuel)$^{-1}$. Emissions from Euro 6 diesel vehicles are in the range of those from petrol vehicles.

Similar trends can be observed for BC emissions of HDVs and buses. The BC EFs of both HDVs and buses drop significantly from Euro III to IV and from Euro IV to V. Measured buses were mainly well maintained city operated Euro V and VI vehicles, with BC emissions in the range of Euro 6 passenger cars.

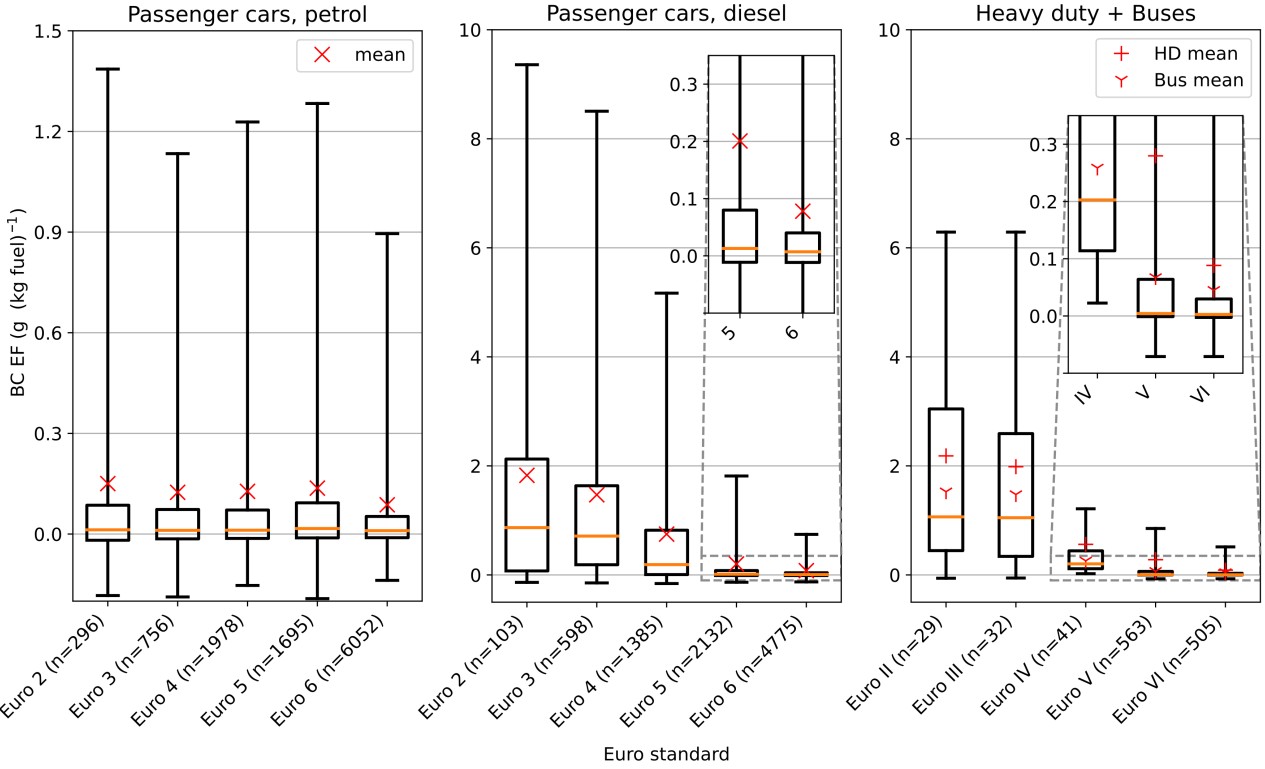

**Figure 15.** Distribution of fuel-based BC EFs in dependence of the Euro emission standard for different vehicle categories. Passenger cars are split into petrol and diesel-powered vehicles. The numbers in brackets represent the sample size.

PN measurements were performed for particles greater than 23 nm (D$_{50}$ cut-off at 23 nm) using a catalytic stripper to remove volatile compounds (Giechaskiel et al., 2014). PN and BC results agree well for the different vehicle categories and

Euro emission standards (Fig. 16). The impact of the introduction of DPF for diesel passenger cars is even more pronounced for PN than for BC. Median PN EFs decrease from Euro 4 to Euro 5 from 320 to $11 \cdot 10^{12}$ particles per kg fuel by a factor of 29. The greater reduction of PN compared to BC EFs can be related to DPF filtration efficiency, which depends on the particle size distribution (Yang et al., 2009; Rossomando et al., 2021). Vehicle exhaust PN consists mainly of a large number of small



particles below 60 nm, while the main contributor to BC mass concentration are accumulation mode particles (Giechaskiel
et al., 2014).

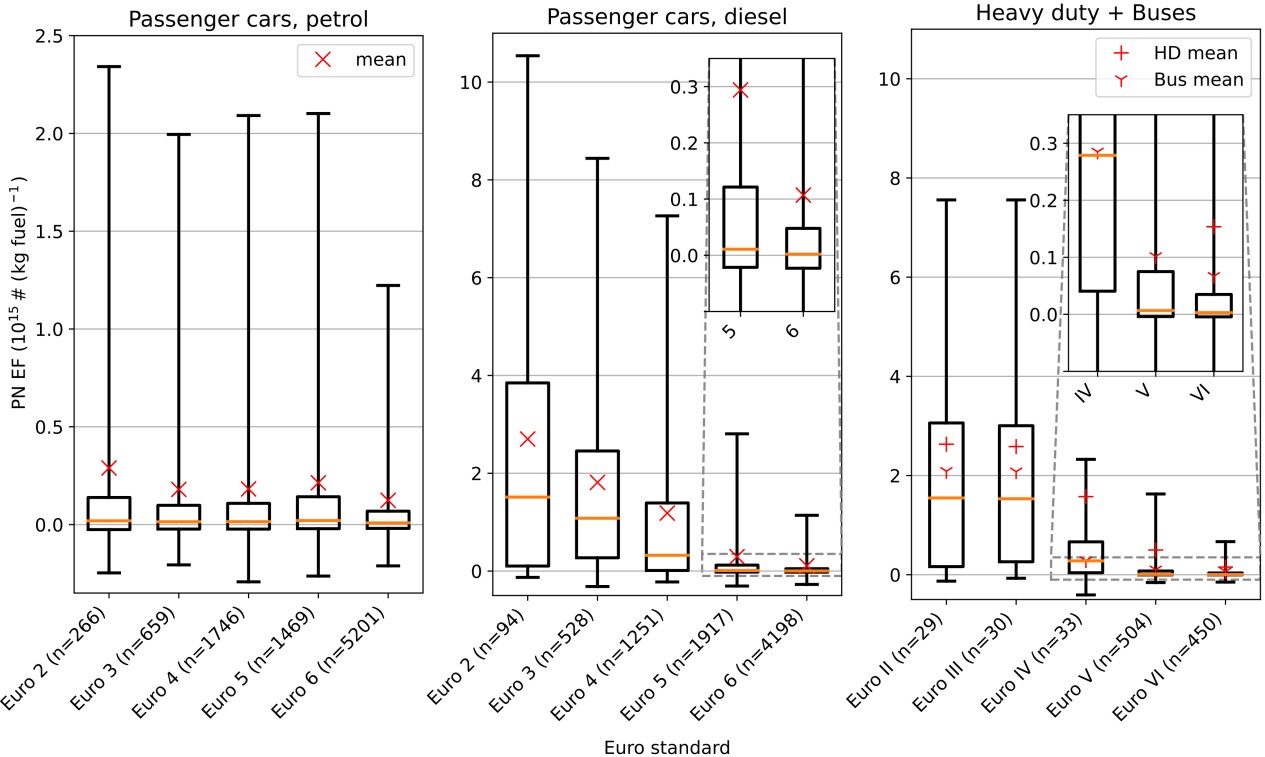

**Figure 16.** Distribution of fuel-based PN EFs in dependence of the Euro emission standard for different vehicle categories. PN measurements
were performed for solid particles greater than 23 nm (SPN$_{23}$). Passenger cars are split into petrol and diesel-powered vehicles. The numbers
in brackets represent the sample size.

NO$_x$ emission levels of petrol passenger cars are steadily decreasing from Euro 2 to Euro 6 (Fig. 17). Median values decrease
from 3.52 g (kg fuel)$^{-1}$ to 0.22 g (kg fuel)$^{-1}$. The effects of "Dieselgate" are reflected in the NO$_x$ emissions of diesel passenger
cars, which primarily affect Euro 5 and Euro 6 vehicles. NO$_x$ EFs stagnate for Euro 2 to 5 vehicles, with median values ranging
from 8.24 to 9.16 g (kg fuel)$^{-1}$. For Euro 6 vehicles, NO$_x$ EFs decrease significantly with a median value of 1.74 g (kg fuel)$^{-1}$.
In contrast to BC and PN, NO$_x$ EFs for HDVs and buses are higher compared to the emissions of passenger cars. This applies
to all Euro classes. HDVs tend to have a higher milage, which affects the deterioration of the vehicle condition. In addition,
intentional tampering of the NO$_x$ reduction system is believed to be more common in commercial vehicles than in private
vehicles.

Table 2 compares average emissions of selected Euro emission standards from this study with previous open-path RES and
PS studies. The average BC EFs from this study are compared with the PM EFs from other studies. BC can be assumed to be a
subset of PM. For diesel vehicles, BC typically accounts for the largest share of PM emissions. This is especially the case for





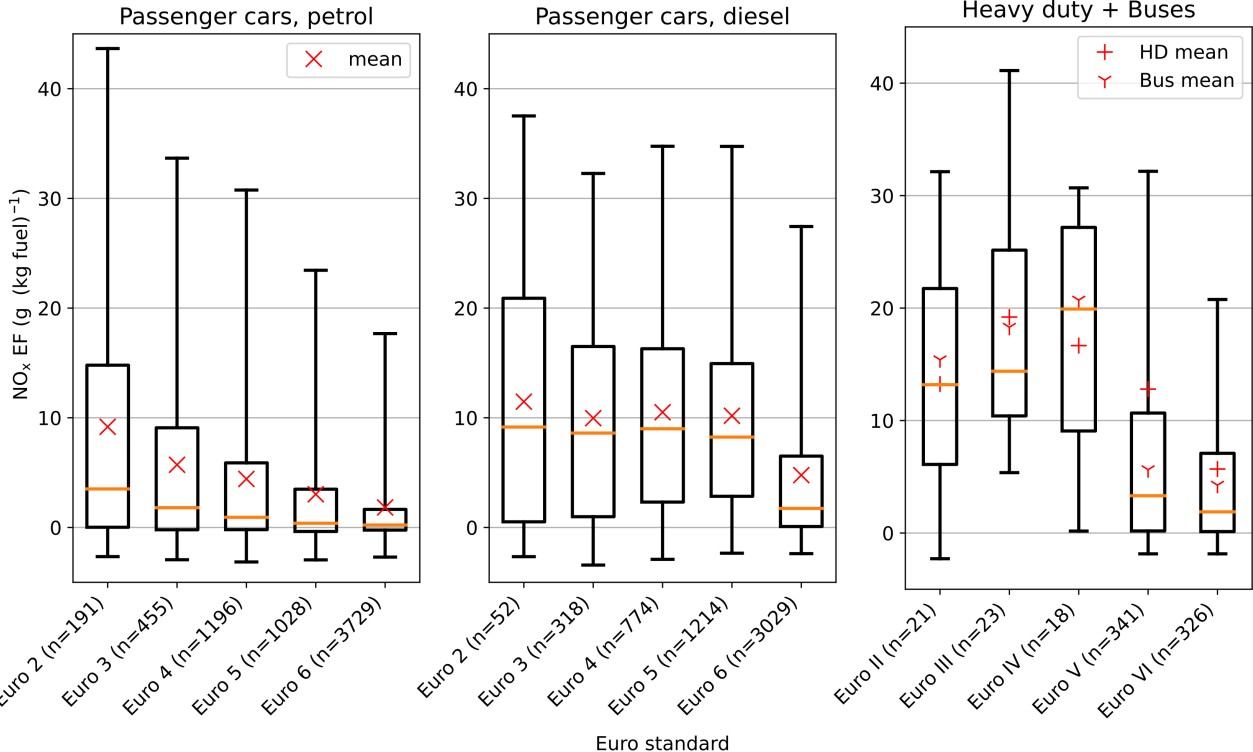

**Figure 17.** Distribution of fuel-based $NO_x$ EFs in dependence of the Euro emission standard for different vehicle categories. Passenger cars are split into petrol and diesel-powered vehicles. The numbers in brackets represent the sample size.

older Euro emission standards and vehicles with defective DPF. The proportion of BC emissions for petrol vehicles is typically lower, except under specific conditions (Platt et al., 2017; Yang et al., 2019; Bessagnet et al., 2022). PN EFs are reported only for PS studies, because only rough estimates from open-path RES studies can be made for PN.

Average BC EFs from this study and PM EFs from open-path studies are in a similar range for petrol and diesel passenger cars. The average PM EFs reported from open-path RES studies are subject to a large variation. The measured emissions can vary widely depending on several factors such as fleet characteristics (e.g. vehicle type, manufacturer, mileage, age) or the measurement location. Determined average BC EFs from this study for Euro 5 and 6 passenger cars are higher than those found in open-path RES studies. One reason for this can be the quantification limit of open-path RES instruments for PM

measurements. Several studies pointed out difficulties in quantifying emissions of newer Euro emission standards, which was reflected in negative average EFs (Gruening et al., 2019; Cha and Sjödin, 2022; Jerksjö et al., 2022). The average $NO_x$ EFs from the selected open-path RES studies vary to a much smaller extent. Determined $NO_x$ EFs for petrol passenger cars of this study agree well with literature values for all Euro emission standards given. $NO_x$ emissions from diesel passenger cars are lower compared to those presented in literature studies. This could be due to peculiarities in the composition of the vehicle

fleet, vehicle updates after "Dieselgate" creating a change in the fleet or due to differences in the measurement principles.





Validation studies (Ropkins et al., 2017; Sjödin et al., 2018; Gruening et al., 2019) show a very good agreement between reference measurements with portable emission measurement systems (PEMS) and open-path RES for NO measurements. The studies found differences in absolute values between PEMS and open-path RES systems, with open-path RES systems partially underestimating NO EFs. Differences could also be due to determined $NO_2$ EFs, where higher inaccuracies were reported for
open-path RES. These could come from concentrations below the quantification limit of open-path RES systems and the volatility of the gas. For PS, detailed validation studies are not available yet. Ježek et al. (2015) reported a good agreement of average BC EFs between PS and PC methods. Larger deviations were reported for the measured PN EFs of the two methods, which could be due to higher uncertainties resulting from PN measurements including volatile compounds.

The calculated EFs of HDVs and buses from this study are compared with selected literature from both PS studies and open-
path RES studies. The selected PS studies were conducted solely in Sweden. BC EFs of this study and PM EFs of literature studies are in similar ranges for Euro III to Euro V standards. Differences are mainly observed for Euro VI HDVs. These can arise from differences in fleet characteristics or vehicle age, causing deterioration of the exhaust after-treatment system. This could be particularly the case for newer Euro VI HDVs, where 2-3 years between the studies can have a significant impact. Negative PM EFs reported by open-path RES studies can be referred to limits in instrument accuracy similar to those for
passenger cars. PN EFs for HDVs and buses reported in previous PS studies are generally higher than those reported in this study. This can be attributed to the different size characteristics of the used PN instruments. Particles larger than 5.6 nm were measured by Hallquist et al. (2013), Liu et al. (2019) and Zhou et al. (2020). In this study, the $D_{50}$ cut-off was 23 nm (Schriefl et al., 2020). We used SPN measurements with a cut-off at 23 nm to comply with current emission regulations and to be able to relate the calculated EFs to official limits. Determined $NO_x$ EFs of HDV and buses are in good agreement with literature data
of PS and open-path RES studies. Similar to diesel passenger cars, $NO_x$ EFs of this study for HDV and buses are slightly lower compared to literature data of open-path RES studies. $NO_x$ EFs of literature PS studies span over a wide range, which can be referred to differences in vehicle exhaust after-treatment systems. In Hallquist et al. (2013), open-path RES data was used to determine $NO_x$ EFs. Which EFs are more accurate can not be judged from the comparison as several reasons influence the derived EFs like measurement location, vehicle fleet, driving properties, environmental conditions, instrument characteristics
or data analysis algorithm. A detailed comparison of our PS system with other simultaneous measurements will be part of a separate study.

## 4  Summary and conclusions

This paper presents a PS system capable of continuously screening entire vehicle fleets. Our approach allows the direct measurement of different particle metrics such as BC or PN as well as different gaseous compounds (e.g. $NO_x$). In particular, PN is
a relevant metric today with knowledge of the health effects of ultrafine particles (Oberdörster et al., 2005; Brook et al., 2010; Mannucci et al., 2015), but also concerning currently introduced emission legislations (Bainschab et al., 2020; Giechaskiel et al., 2021). Newly introduced Euro emission standards bring along stringent requirements, where current open-path RES systems reach their quantification limits, especially for PM (Gruening et al., 2019; Cha and Sjödin, 2022; Jerksjö et al., 2022).





| Study | Vehicle type - fuel type | Euro standard | Av. EF BC/PM mg (kg fuel)$^{-1}$ | Av. EF PN $10^{12}$ # (kg fuel)$^{-1}$ | Av. EF NO$_x$ g (kg fuel)$^{-1}$ |
|---|---|---|---|---|---|
| This study | Passenger cars - petrol | Euro 3 | 124[a] | 180[c] | 5.7 |
| Open-path RES studies[1,2,3,4] | | | 30 - 670[b] | - | 5.1 - 14.5 |
| This study | | Euro 4 | 127[a] | 182[c] | 4.4 |
| Open-path RES studies[1,2,3,4] | | | 20 - 200[b] | - | 3.8 - 7.2 |
| This study | | Euro 5 | 137[a] | 214[c] | 3.0 |
| Open-path RES studies[1,2,3,4] | | | 30 - 90[b] | - | 2.5 - 3.3 |
| This study | | Euro 6 | 97[a] | 139[c] | 2.1 |
| Open-path RES studies[1,2,3,4] | | | 0 - 95[b] | - | -0.5 - 3.2 |
| This study | Passenger cars - diesel | Euro 3 | 1470[a] | 1813[c] | 10.0 |
| Open-path RES studies[1,2,3,4] | | | 170 - 1840[b] | - | 13.7 - 18.7 |
| This study | | Euro 4 | 749[a] | 1185[c] | 10.5 |
| Open-path RES studies[1,2,3,4] | | | 130 - 1080[b] | - | 11.6 - 15.4 |
| This study | | Euro 5 | 201[a] | 294[c] | 10.2 |
| Open-path RES studies[1,2,3,4] | | | 20 - 270[b] | - | 11.7 - 14.4 |
| This study | | Euro 6 | 80[a] | 116[c] | 6.8 |
| Open-path RES studies[1,2,3,4] | | | 10 - 70[b] | - | 5.8 - 8.5 |
| This study | Buses, heavy duty - diesel | Euro III | 1476, 1985[a] | 2078, 2585[c] | 18.3, 19.2 |
| PS studies[6,7,8] | | | 30 - 1820[b] | 730 - 3900[d] | 16 - 43.3[e] |
| Open-path RES studies[1,4] | | | 250 - 2100[b] | - | 24.6 - 27.5 |
| This study | | Euro IV | 259, 561[a] | 285, 1575[c] | 20.7, 16.7 |
| PS studies[6,8] | | | 172 - 1845[b] | 870 - 3200[d] | 14 - 19.8[e] |
| Open-path RES studies[1,4,5] | | | 220 - 1250[b] | - | 17.8 - 21.5 |
| This study | | Euro V | 67, 280[a] | 102, 498[c] | 5.6, 12.8 |
| PS studies[6,7,8] | | | 146 - 258[b] | 650 - 1600[d] | 15 - 37[e] |
| Open-path RES studies[1,2,4,5] | | | 40 - 360[b] | - | 13.1 - 25.3 |
| This study | | Euro VI | 45, 88[a] | 67, 154[c] | 4.3, 5.7 |
| PS studies[8] | | | 5[b] | 850[d] | 3.1 |
| Open-path RES studies[1,2,4,5] | | | -50 - 190[b] | - | 2.8 - 8.7 |

**Table 2.** Comparison of average fuel-based BC/PM, PN and NO$_x$ EFs of this study with selected open-path RES and PS literature data. Emissions are ordered by vehicle category, fuel type and Euro emission standard. BC/PM column show either BC or PM EFs. [1]Hooftman et al. (2019), [2]Bernard et al. (2021), [3]Jerksjö et al. (2022), [4]Cha and Sjödin (2022), [5]Lee et al. (2022), [6]Hallquist et al. (2013), [7]Liu et al. (2019), [8]Zhou et al. (2020), [a]BC, [b]PM, [c]PN > 23 nm, [d]PN > 5.6 nm, [e]NO EFs of Hallquist et al. (2013) were determined by open-path RES, NO$_2$ measurements are estimated.




Compared to commercial open-path RES systems, the installation of the measurement setup is relatively simple. The method is quite flexible when it comes to where the sample extraction is performed and what instruments are used to measure the species of interest. We presented a comprehensive data analysis framework that is capable of processing emissions from 1,000s of vehicles. This encompasses the pre-processing of raw time series of instruments to the post-analysis of emission statistics of whole vehicle fleets. The core of this software is the TUG-PDA, which determines and separates vehicle emissions down to a distance of 3 s between the vehicles, if appropriate instruments are used. The data analysis provides the capability to analyze emissions for different vehicle characteristics such as emission standard, age, mileage or manufacturer. This allows detailed analysis of emission patterns to investigate whether emission standards are being met and also helps to identify the causes of high emissions. As an application example, we presented the first results of measurement campaigns in three different European cities in which we made use of our PS method. We showed distributions of measured BC, PN and $NO_x$ EFs of different vehicle types and Euro emission standards. The results are in good agreement with relevant literature studies and showed the potential for screening emissions of vehicles with newly introduced emission standards.

We evaluated important impact factors influencing PS measurements, which should be considered when planning and implementing PS campaigns. As our evaluations show, the CR can vary greatly depending on the different factors. The most important influences are summarized in the following:

– Instruments used for PS campaigns should be properly chosen (see Table 1). Response time, dynamic range and LoD are the most significant factors. The response time should be as low as possible ($< 1$-2 s). The dynamic range should be large enough to cover both low and high emitters. The LoD should be low enough to accurately determine the emissions from the evaluated species.

– When selecting the measurement site, the traffic conditions must be taken into account. An ideal condition is a steady traffic flow with sufficient distance ($\geq 3$ s with appropriate instrumentation) between the vehicles to collect a high number of valid emission records. At the measurement site, the passing vehicles should be under considerable engine load. This can either be in appropriate traffic situations where vehicles accelerate (e.g., after crossing, slip road) or at roads with positive gradients.

– The sampling should be performed as close to the exhaust source (tail pipe) as possible. The best results can be achieved if the sample extraction is performed from the middle of the road with the sampling tube directly attached to the road. When sampling from the side, the road width (smaller roads preferred) and the sampling height (as low as possible) have a significant influence on the CR. In addition, the position of the exhaust pipe of the fleet of interest should be examined in advance to determine the best sampling position.

– The CR depends on the wind speed and direction. Windy conditions that transport the exhaust plume away from the sampling point have a negative effect on the CR. The influence of wind can be minimized by choosing an appropriate measurement location (e.g., street canyon). Other weather factors such as temperature or precipitation have negligible impact.



- The impact of harsh and varying BG conditions on the instruments, as well as on the data analysis, must be considered. Local (directly before the vehicle pass) BG concentrations should be used to minimize calculation errors. Differences in the BG determination have a large influence on the resulting EFs for low emitters and highly diluted plumes.

590
- Care should be taken to align the measured time series data. Misalignments between the response time (i.e. between the vehicle pass time and instruments) or between sensors ($CO_2$ and pollutants) can cause substantial errors. This can already be the case for time shifts of 1-2 s.

Future work will involve a detailed analysis of the gathered BC, PN, and $NO_x$ emissions. Several open questions will be addressed such as how PS measurements relate to those made by reference equipment such as PEMS or other methods. Another
595 interesting aspect to be investigated is the reproducibility and reliability of individual measurements. This is particularly important for the potential identification of high emitters. Commercial open-path RES systems are associated with a high degree of uncertainty when considering individual measurements (Huang et al., 2020; Qiu and Borken-Kleefeld, 2022). PS results indicate that single measurements are more reliable due to the longer measurement duration. It would be a great advance if this could be proven and applied in the future to identify individual high emitters.





## Appendix A: Remote emission sensing definitions

### A1  Emission ratio and fuel-based emission factor

Emissions of combustion-based vehicles are generally reported as EFs. In RES, fuel-based EFs are used to express emissions from the measured vehicles (Hansen and Rosen, 1990; Borken-Kleefeld et al., 2018; Bernard et al., 2018). Fuel-based emissions are expressed as a mass fraction of the emitted pollutant per mass of burned fuel. The amount of burned fuel is calculated based on the measured $CO_2$ concentration of the passing vehicle by using the carbon mass balance method (Bishop et al., 1989; Hansen and Rosen, 1990; Stedman et al., 1992; Singer and Harley, 1996; Ban-Weiss et al., 2009; Hak et al., 2009) and under the assumption that the majority ($> 90$ %) of the carbon content in the fuel is oxidized to $CO_2$ during the combustion process. By relating the measured pollutant P (e.g., BC, PN, $NO_x$) to the measured $CO_2$ concentration, an ER can first be calculated (Stedman et al., 1992; Hansen and Rosen, 1990):

$$ER = \frac{\int_{t_1}^{t_2}([P]_t - [P]_{t_0})dt}{\int_{t_1}^{t_2}([CO_2]_t - [CO_2]_{t_0})dt} \tag{A1}$$

By multiplying the ER with the mass fraction of carbon in fuel $\omega_c$ a fuel-based emission factor $EF_{fb}$ (Ban-Weiss et al., 2009; Hak et al., 2009) can be calculated:

$$EF_{fb} = ER \cdot \omega_c, \tag{A2}$$

where $\omega_c$ discriminates among different fuel types (see Table A1). Regarding PS measurements, the emission events last for several seconds. The measured concentrations of the pollutant and $CO_2$ for one vehicle pass are commonly integrated and the results are related to each other. The start and stop times of the emission event define the integration intervals and these are represented by $t_1$ and $t_2$, respectively. The measured emissions are superimposed by BG concentrations of the different species in ambient air. For that reason, a BG correction is required, where $t_0$ specifies the point of time from which the BG concentration is used. The BG is usually determined on the basis of the concentration at the integration starting point, $t_1$. When plumes overlap or impacts from other sources occur, this concentration may be underestimated.

| Fuel type | $\omega_c$ |
|---|---|
| Gasoline (2016 E0) | 0.864 |
| Diesel (B0) | 0.861 |
| CNG | 0.708-0.717 |
| LPG | 0.824 |
| LNG | 0.749-0.756 |

**Table A1.** Typical mass fraction of carbon in common fuel types from JRC (2020).





## A2 Distance-based emission factor

Fuel-based EFs do not distinguish between vehicles with different fuel consumption. Therefore, fuel-based EFs favor vehicles with higher fuel consumption. Distance-based EFs, on the contrary, include the fuel consumption and are, therefore, usually used to compare vehicle emissions (Bernard et al., 2018). However, distance-based EFs cannot be directly calculated in RES

due to the snapshot measurement. An estimate can be calculated using the type approval $CO_2$ consumption ($Avg_{CO_2}$) which is obtained from the vehicle technical information. The official $CO_2$ emissions from passenger cars have been shown to differ increasingly from real-world emissions (Tietge et al., 2017). Therefore, a correction factor ($RWG_{CO_2}$) including the real-world $CO_2$ gap can be included according to Bernard et al. (2018). Making this assumption, a distance-based EF can be calculated as follows:

$$EF_{db} = ER \cdot Avg_{CO_2} \cdot RWG_{CO_2} \tag{A3}$$

In a recent publication by Davison et al. (2020), a new approach was presented to calculate the distance-based EF in RES studies using the VSP. A good agreement was found between the outcome of their approach and validation measurements made with portable emission measurement systems (PEMS).

## A3 Vehicle specific power

VSP is often used when performing emission modeling of combustion-based vehicles to estimate vehicle operating conditions. Using VSP, insights can be gained to estimate the engine load when a vehicle passes the measurement point. For example, this can be used to exclude vehicles with a small VSP ($< -5$ kW t$^{-1}$) due to the disabled fuel injection in the engine (and therefore unexpected $CO_2$ emissions) (Bernard et al., 2018). The VSP is defined according to Jimenez-Palacios (1999) by the sum of the relevant power variables related to the mass of the vehicle:

$$VSP = \frac{\frac{d}{dt}(E_{Kin} + E_{Pot}) + F_{Rol} \cdot v + F_{Aero} \cdot v}{m} = v \cdot (a \cdot (1 + \epsilon) + g \cdot grade + g \cdot C_R + \frac{1}{2} \cdot \rho_a \cdot C_D \frac{A}{m} \cdot (v + v_w)^2) \tag{A4}$$

Where $E_{Kin}$ and $E_{Pot}$ are the kinetic and the potential energies, respectively, $v$ is the speed of the vehicle, $F_{Rol}$ is the force from the rolling friction, $F_{Aero}$ is the aerodynamic drag force, $m$ is the mass of the vehicle, $a$ is the vehicle acceleration, $\epsilon$ is the mass factor, $g$ is the gravity of Earth, $grade$ is the road gradient, $C_R$ is the coefficient of the rolling resistance, $\rho_a$ is the density of ambient air, $C_D$ is the drag coefficient, $A$ is the projected frontal area of the vehicle and $v_w$ is the headwind impacting the

vehicle.

## Appendix B: PS emission measurement setup

Fig. B1 shows the emission measurement setup used during one of the measurement campaigns. A water protection hose protected the sampling inlets from water. Two different sampling tubes were used for particle and gas sampling. Tygon tubing



with a inner diameter of 5 mm was used for particle sampling. Teflon tubes were used for the $NO_x$ measurements, due to the
volatility of the gas. Both sampling paths were protected with water traps. For the particle measurements, impactors removed
particles with diameters greater than 1 $\mu$m. BC and $CO_2$ were measured with the newly developed BCT (Knoll et al., 2021).
In parallel measured an Aethalometer AE33 (Magee Scientific) BC and brown carbon (BrC). Total particle number (TPN)
concentration was measured using a CPC (Condensation Particle Counter 3775, TSI Incorporated). SPN measurements were
conducted with a diffusion charger (Schriefl et al., 2020) downstream to a custom-built catalytic stripper which removed volatile
compounds. A second BCT measured in parallel to the diffusion charger BC and $CO_2$. A dust filter protected the instrument in
the gas sampling path from particle penetration. $NO_x$ and $CO_2$ were measured with an ICAD (Airyx GmbH, Horbanski et al.
(2019)).

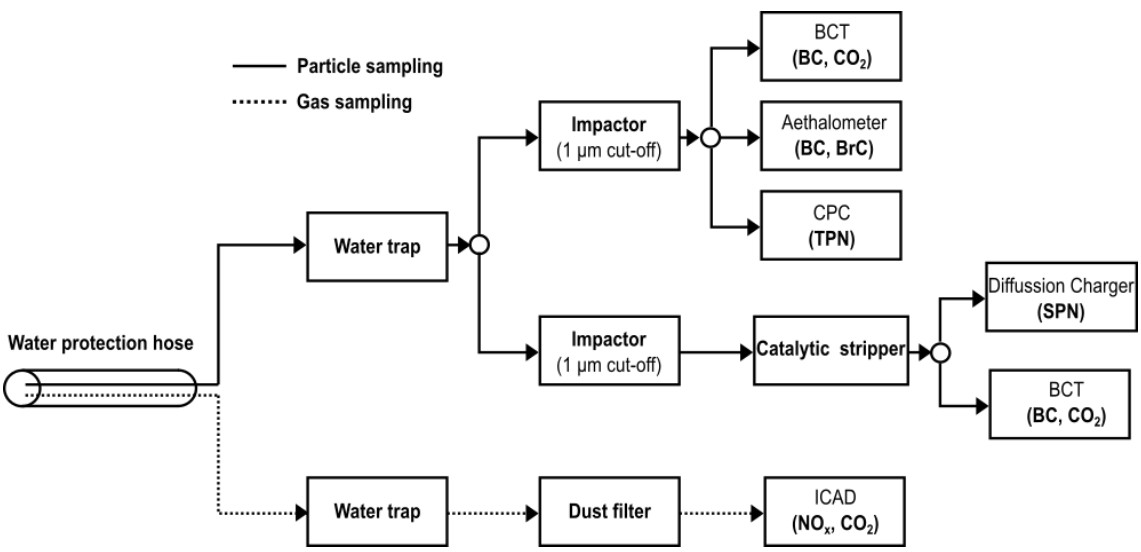

**Figure B1.** PS emission measurement setup used during one of the campaigns.

**Appendix C: TUG-PDA emission separation capabilities**

Fig. C1a and Fig. C1b show two PS time series examples that demonstrate the capabilities and the limits of the TUG-PDA
for plume separation of passing vehicles. Four vehicles passed by the PS spot, as shown in the data presented in Fig. C1a.
The TUG-PDA detects the rising concentrations (increasing gradient) of BC and $CO_2$ when the first vehicle (V1) passes and
integrates the areas (A1) until the determined $CO_2$ BG concentration is undercut. When the second vehicle passed (V2), a
much weaker plume was captured and only a minor increase in both $CO_2$ and BC (A2) can be detected. The mean value of the
turning point from negative to positive gradient, and a concentration value from the last time window where no vehicle passed
is used as BG concentration. Four seconds after the second passage, a third vehicle (V3) passed by. A substantial increase in
both the BC and $CO_2$ concentrations could be measured. The plumes could be properly separated by the TUG-PDA as marked



by the integrated areas (A3). Afterwards, another vehicle passed by (V4). The emissions assigned to the fourth vehicle can be easily determined by the TUG-PDA and are also highlighted (A4). In this example, the emissions of the passing vehicles can be properly resolved.

In contrast, Fig. C1b shows a data example where the emissions of most passing vehicles cannot be properly resolved. The first vehicle (V1) passes by, and small increase in $CO_2$ (and no BC emissions) can be seen. The exhaust plume is integrated by the TUG-PDA (A1). The plume of the second passing vehicle (V2) can also be properly detected (A2). During the integration of the emissions of V2, a third vehicle passes the PS site, but no change in $CO_2$ or BC concentrations is measured. Therefore, the TUG-PDA continues the processing of the exhaust plume for V2. For V3, no EF can be determined. Four seconds later, two

vehicles (V4, V5) pass the PS spot with a gap of just one second. A significant increase in the $CO_2$ concentration is measured after V4 passed by. After a small dip, the $CO_2$ concentration rises again, accompanied by a significant BC peak which are probably caused by V5. The emissions from the two vehicles are superimposed, and therefore cannot be certainly assigned to one of the two passing vehicles with such a short distance between them. Six seconds later, vehicle V6 is detected by the light barriers. A slight increase in both $CO_2$ and BC concentrations can be seen but the captured $CO_2$ peak is too small. After

a short gap of nine seconds V7 is passing by. With a small delay of around 2 seconds, a strong $CO_2$ plume is measured along with a small BC peak. During the increase in both concentrations vehicle V8 passed the measurement point. EFs cannot be correctly determined for neither V7 or V8 due to the delay of the measured emissions from V7 and the short distance between the vehicles. Two seconds after V8, two vehicles (V9, V10) pass by. An increase in $CO_2$ can be seen that is overlaid by the previous emissions from V7. No EFs can be determined for V9 and V10 because of the small spacing between them. Mainly

two criteria in the TUG-PDA are responsible if emissions cannot be properly separated. First, a minimum spacing of three seconds between the vehicles is specified for the TUG-PDA. If vehicles pass by with a spacing of less than three seconds, the emissions cannot be certainly attributed to one passing vehicle. Second, if the emission concentration is steadily rising over multiple vehicle passes the determined emission can not be correctly assigned to one vehicle.

## Appendix D: Influence of weather conditions

Fig. 11b and Fig. D2 show the influence of wind direction on the CR. We separated the directions in the 180° when the wind blows the plume towards the sampling point and in the 180° when the wind blows the plume away from the sampling point (Fig. D1).

The impacts of wind direction and speed on the capture rate at urban sampling locations are shown in Fig. D2. No real difference can be seen between winds blowing the exhaust towards or away from the sampling location. We assume that this

is mainly due to differences between the measured local wind conditions at the sampling location and the measured data from the weather station nearby.



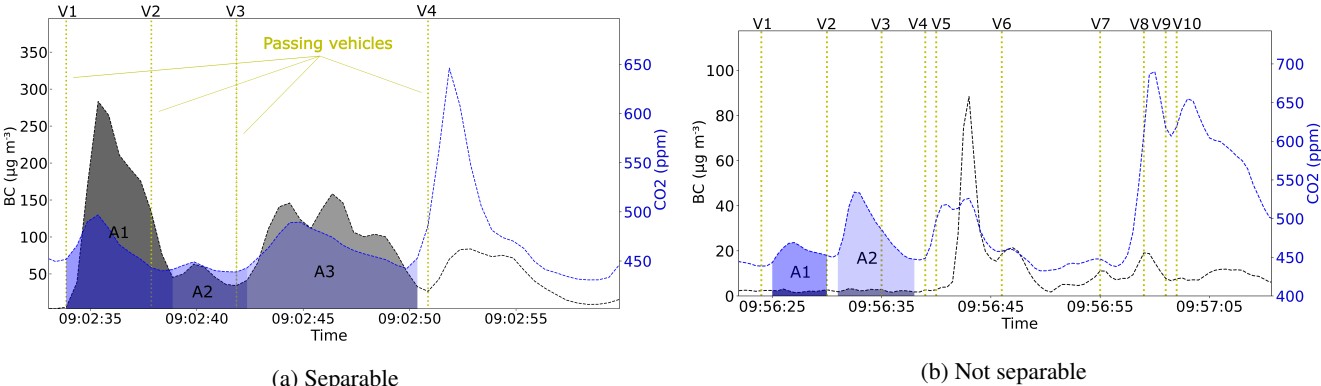

(a) Separable

(b) Not separable

**Figure C1.** Two PS time series examples (BC, $CO_2$) from captured plumes. The yellow, vertical, dashed lines mark the point of time when the vehicles passed the PS spot. **a)** Emissions can be separated for the individual vehicle passes. Four vehicle passes are shown where the assigned emissions are highlighted in different color schemes (A1 - A4). **b)** For most vehicle passes, the emissions for individual vehicle passes are not-separable. Ten vehicle passes are shown, of which emissions can be correctly determined for two.

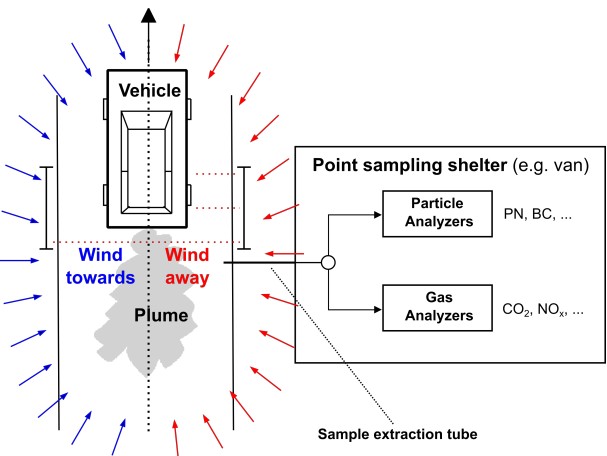

**Figure D1.** Schematic of the PS setup where the sampling is done from the roadside. The wind directions are indicated for winds which blow the exhaust plume toward or away from the sampling point.

## Appendix E:  Influence of misaligned measurement data

We emphasized the importance of compensating for sampling delays and the response times of instruments during the pre-processing steps of the data analysis. Additionally, the instrument responses are time-aligned to the vehicle passes which cause the fastest response of the measurement instruments (see Sect. 2.2.1). Misalignment has a major influence and impacts both the CR and the resulting ERs. Therefore, we investigate the effects resulting from misalignment between emission datasets from different instruments (e.g., $CO_2$ and pollutants) and between emission data and the vehicle pass times. First, we take a look at the latter. We deliberately misaligned the time series of the measurement equipment (e.g., $CO_2$, BC) as compared to the vehicle






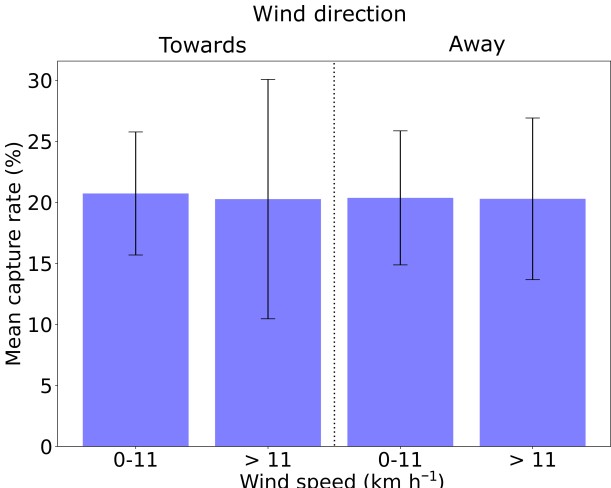

**Figure D2.** Influence of wind speed and direction in urban areas on the capture rate.

pass times for time shifts (misalignment) between -5 and +5 s. Negative time shifts represent exhaust plumes that occur prior

to the corresponding vehicle pass. A maximum CR is reached at a time shift of -1.5 s (Fig. E1a). There are several reasons why

the CR peaks at slightly misaligned data. First, the TUG-PDA starts to scan the emission concentrations 1 s before the vehicle

pass time (see Fig. 4). Second, the emission concentration time series datasets are time-aligned for plumes with the smallest

time delay. These are plumes, e.g., of vehicles with the exhaust pipe located on the same side as the sampling position (if the

sampling is performed from the side). Plumes of vehicles with an exhaust pipe on the other side are sampled with a slight delay

(up to a few seconds). The negative time shift leads to additionally captured plumes in cases where the vehicles pass within

small distance of each other. These additional records can be both correctly or wrongly assigned emissions, depending on the

circumstances. For prior perfectly aligned plumes parts of the emissions are cut off because they are outside of the TUG-PDA

start range leading to deviations in the resulting ERs. As the misalignment increases, the CR decreases steadily. The main

reason is that the plumes from vehicles passing within short distances of one another can no longer be resolved. The substantial

drop in time shifts below -3 s is caused by plumes peaking before the vehicles pass (e.g, see Fig. E5), and these peaks are

not detected for larger negative time shifts. In addition to the influence on the CR, misalignment has a major influence on the

resulting ERs. Fig. E1b shows the deviation of the calculated ERs caused by misalignment compared to the results calculated

with the aligned datasets. The deviation increases as the misalignment increases, with median values ranging from -31.4 % to

0 % and between -162.1 % and 242.6 % on average. Fig. E2 shows the impact of misalignment between vehicle pass times

and measured emission data on the integrated $CO_2$ concentration. The $CO_2$ concentration increases as the positive time shift

increases and decreases as the negative time shift decreases. An increasing negative time shift in the time series causes the

plumes to be cut off. Thus, BG concentrations that are too high are determined, which reduce the integrated concentrations

(see e.g. Fig. E4). The deviations of the misaligned, integrated $CO_2$ concentrations from the properly aligned data are shown

in Fig. E3.



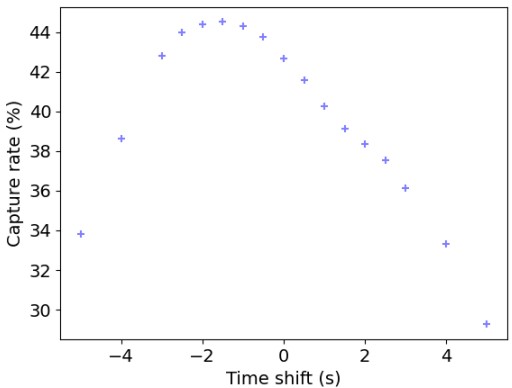

(a) Influence of misalignment on the capture rate.

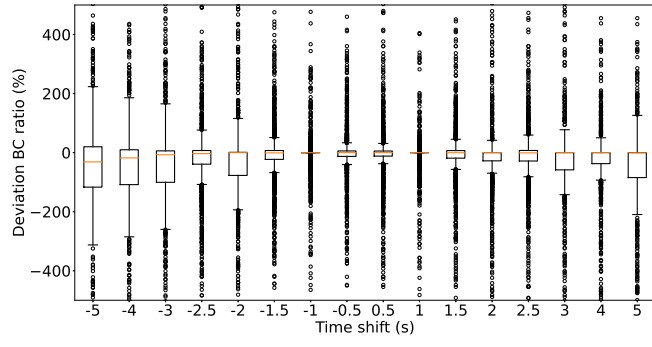

(b) The deviation of the resulting BC emission ratio due to the misalignment compared to the properly aligned data is shown.

**Figure E1.** Impact of improperly compensated sampling delay due to misalignment of time series data of the measurement equipment with the vehicle pass times.

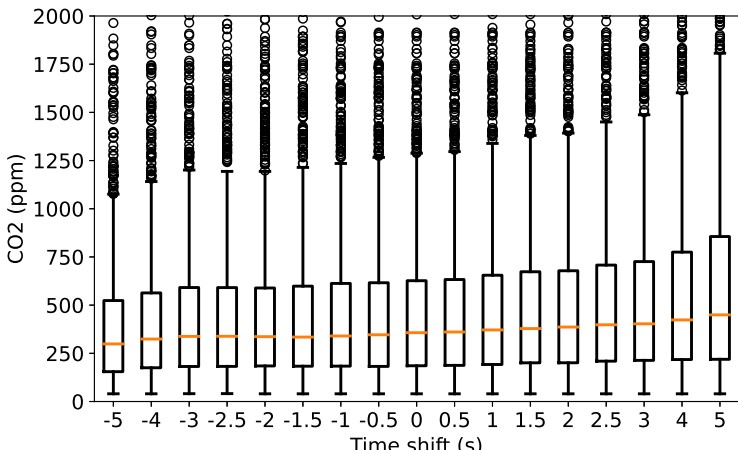

**Figure E2.** Impact of improperly compensated sampling delay on the integrated $CO_2$ concentration of the passing vehicles.

Two examples of misaligned instrument data related to the vehicle pass times are depicted in Fig. E4 and Fig. E5. The emission concentration time series from the instruments are 3 s time shifted (misaligned on purpose) compared to the vehicle pass times for both cases. In the graphs only the $CO_2$ concentration is shown. In both examples, the aligned data is shown in the same graph as the misaligned data. The TUG-PDA start range is shown on top, to illustrate the time range in which the TUG-PDA searches for an increasing concentration. The first case (Fig. E4) shows an example where the time shift can be compensated by the TUG-PDA. No other vehicle interferes with the measurement and the captured emissions are not too much delayed for proper detection. Only a small amount of the emissions are missed (highlighted in red). The second example





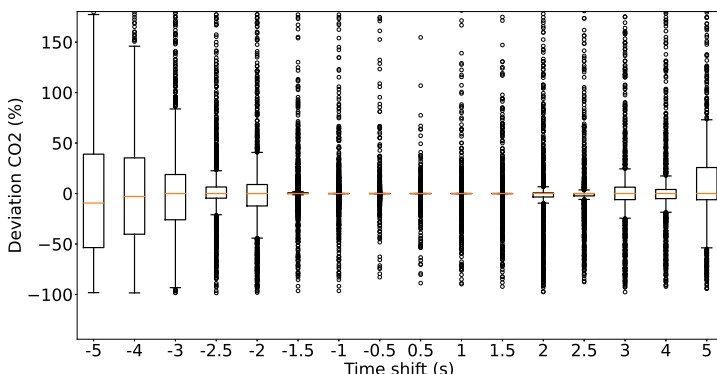

**Figure E3.** Impact of misalignment between measured emission concentrations and the vehicle pass times. Deviation of the properly aligned integrated $CO_2$ signal compared to the time shifted signal.

(Fig. E5) shows one case where the emission concentrations are shifted to such an extent that they can no longer be associated with the vehicle pass. The positive slope of the plume is not covered by the TUG-PDA start range and therefore the entire plume is missed.

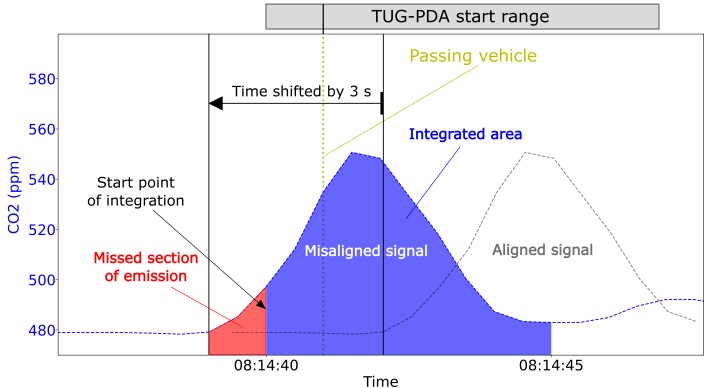

**Figure E4.** Time series example where the misalignment between instrument data (only $CO_2$) and vehicle pass times can be compensated by the TUG-PDA. The misaligned as well as the properly aligned time series are shown in the same plot.

As a second step we analyzed the impact of misalignment between instrument data. In particular, $CO_2$ time series data must be properly aligned with respect to data from the measured pollutants (e.g., BC, PN, $NO_x$). We shifted the $CO_2$ time series in 0.5 s steps against the BC time series to investigate possible effects of misalignment on the results. Fig. E6 shows the deviation of the BC ratio as compared to the properly aligned data. As the positive time shift increases, there is a steady increase in the deviation. The deviation increases on average from -0.9 % for +0.5 s to -89.1 % for +5 s. A negative time shift has a different





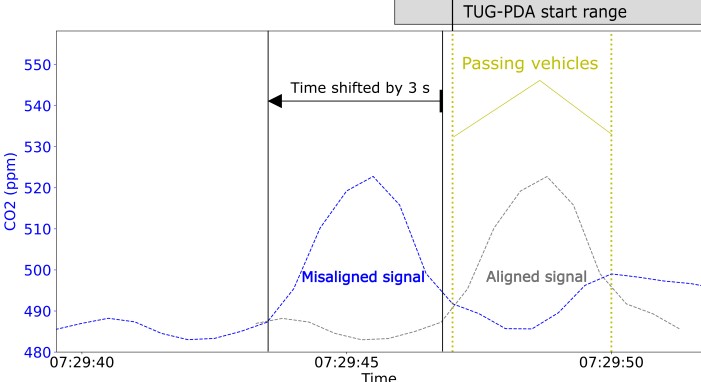

**Figure E5.** Time series example where the misalignment between instrument data (only $CO_2$) and vehicle pass times cannot be compensated by the TUG-PDA. The misaligned as well as the properly aligned time series are shown in the same plot. The TUG-PDA start range is shown for the first vehicle pass.

impact. Statistically, a small and slightly increasing impact is observed up to a time shift of -2.5 s, with a maximum median deviation of 6.4 %. From a time shift of -3 s and onward, the deviation increases significantly, with median values falling between -64.5 % and -89.1 %. The small impact for time shifts between -2.5 and 0.5 s is mainly due to the peculiarities of the TUG-PDA. A small time shift between $CO_2$ and the pollutant is compensated, especially in the negative direction, because the datasets are processed individually. This effect is even more pronounced in the negative direction, because the data are

time-aligned for vehicle plumes with the smallest time delay as already described above. The high deviations observed for small time shifts can mainly be referred to vehicle passages in a short period, where small misalignment errors have a strong impact on the determined ERs.

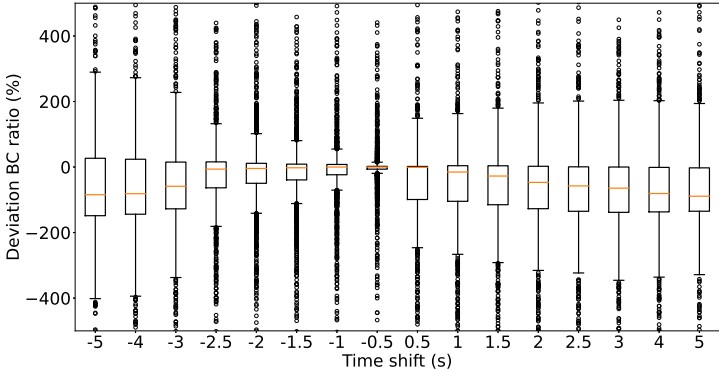

**Figure E6.** Impact of misalignment between $CO_2$ and BC instrumentation. The deviation caused by the misalignment of the resulting BC ER compared to the properly aligned data is shown.



In Fig. E7 and Fig. E8, two time series examples show possible effects of misalignment between the $CO_2$ and BC sensors. In these two cases, the $CO_2$ response is 3 s time shifted as compared to the BC response. In both examples, the original aligned data in the left graph is compared to the misaligned data in the right graph. In the first example (Fig. E7), the misalignment can be compensated by the TUG-PDA for two reasons. First, the emissions from the passing vehicle reach the sample inlet about 3 s after the vehicle has passed by. This could be the case because the tailpipe of the vehicle is located on the left-hand side and the sampling is conducted on the right-hand side, resulting in a transport delay. Second, the whole plume can be captured because there is no other vehicle present, which could cause an interfering plume. The second example (Fig. E8) shows one case where the time shift causes emissions to be assigned to the wrong vehicle. A substantial share of the $CO_2$ emissions of the second vehicle are wrongly assigned to the first vehicle. This results into a underestimated ER for the first vehicle.

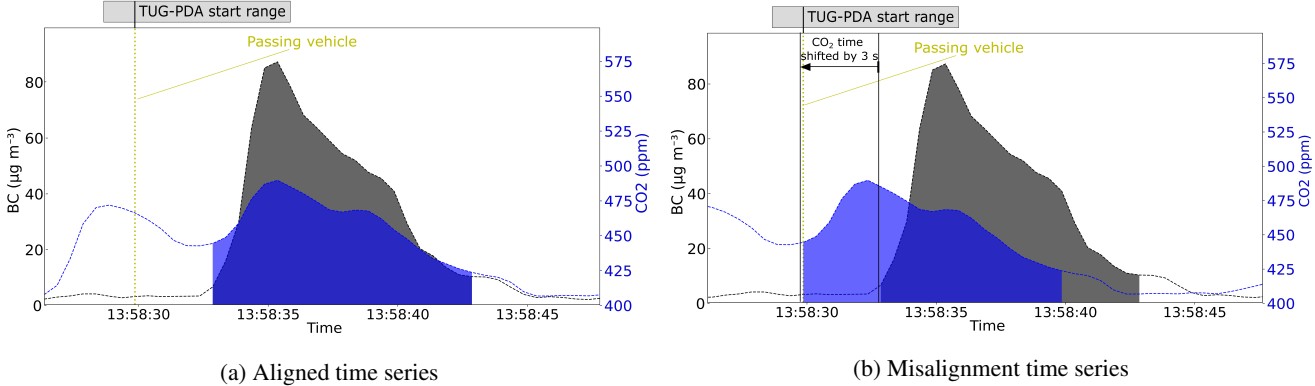

(a) Aligned time series        (b) Misalignment time series

**Figure E7.** Time series example where the misalignment between $CO_2$ and BC sensors can be compensated by the TUG-PDA. Gray (BC) and blue ($CO_2$) shaded areas show the integrated areas.

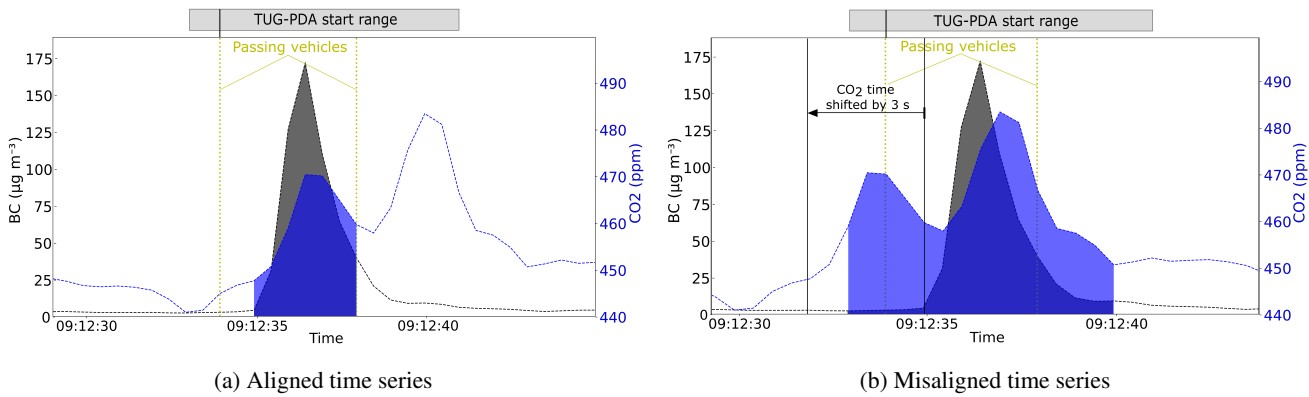

(a) Aligned time series        (b) Misaligned time series

**Figure E8.** Time series example where the misalignment between $CO_2$ and BC sensors cannot be compensated. Gray (BC) and blue ($CO_2$) shaded areas show the integrated areas for the emission concentrations of the first passing vehicle. The TUG-PDA start range is shown for the first vehicle pass.



*Code availability.* On request to the authors.

*Data availability.* Measurement data is available in the CARES database for the PS measurements and on request to the authors.

*Author contributions.* The conceptualization was done by MK and AB. The measurements were conducted by MK, HJ and CS. The method-
ology and investigations were completed by MK, MP, HJ, CS, DP and AB. The software and material preparation were done by MK. MK
drafted the original manuscript, which was reviewed by MP, HJ, CS, DP and AB. The project was supervised at Graz University of Technol-
ogy by AB.

*Competing interests.* The authors declare that they have no conflict of interest.

*Acknowledgements.* This work is supported by TU Graz Open Access Publishing Fund. This work received funding from the EU H2020
project CARES (grant agreement No. 814966). We want to thank Åke Sjödin (IVL) for the great support during the whole project. We
thank Yoann Bernard (ICCT) for the organization of the overall campaigns. We want to thank Åsa Hallquist (IVL) for the support with the
PS measurements. We thank TNO for the support during the characterization experiments and for providing an ANPR camera system for
all measurement campaigns. We thank David Carslaw and Naomi Farren for their support during the project. Many thanks to Innovhub and
AMAT for the organization and their help during the Milan campaign. We want to thank ZTP Krakow and Bartosz Piłat for the support during
the Krakow campaign. Many thanks to Michal Vojtíšek and Martin Pechout for their help during the Krakow and Prague campaigns. Finally,
we want to thank Sitaram Stepponat and Andreas Steiner for their continuous support during the project, especially for the preparation of the
measurement campaigns.



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
