# Peer review of "Large-scale automated emission measurement of individual vehicles with point sampling"

_EGUsphere, 2023_

## Author Response (AR1)

We gratefully thank the reviewers for carefully reading and providing feedback to our manuscript. Below we provide our point-to-point responses to the reviewer's comments. The comments by the reviewer are marked in **black**, responses are marked in **red** and changes to the manuscript are indicated in **blue**.

**Referee 1**

This study developed and demonstrated a point sampling method to automatically measure emissions from a large-scale of individual vehicles. In this works, the authors present their system that can be used for particulate matter (PM) and gas emissions measurements, which is notably independent of vehicle type. They find that when using their peak detection algorithm (TUG-PDA), they can separate vehicle-specific emissions down to a spacing of just a few seconds between vehicles. In this study, they present initial findings from the use of this method that collected ~100,000 vehicle records from several measurement locations, mainly in urban areas. Their findings include a detailed evaluation of the main influencing factors on point sampling measurements, specifically for carbon dioxide ($CO_2$) and black carbon (BC) measurements. When compared to equivalent remote sensing measurements, the authors found good agreement even with the newest standards which are harder to capture due to their lower emissions and the current remote sensing abilities. This paper is well written and organized.

However, the novelty of such work needs to be explored further. While the authors specify that this point sampling method is novel and/or surpasses the ability of many other (cited) studies on roadside emission measurement, this reviewer questions that assumption with the only notable differences coming in the use on light duty vehicles and the automation. It needs to be clear how this work contributes to the scientific knowledge on vehicle emissions measurements.

We thank the reviewer for the comments and suggestions. We summarized below the novelty of this work as we think there is more than "only" light duty vehicles and the automation:

1) We presented a software framework and a peak detection algorithm for automatic emission post-processing that is capable of delivering emission factors for thousands of vehicles. Comparable software frameworks have not been published in literature yet. In addition, we show that the algorithm is able to separate plumes, allowing measurements in dense traffic.

2) Until yet, very limited particulate matter data of real driving emissions of light duty vehicles were published. In Europe, especially particle number emissions are nowadays of interest due to the latest introduced emission standards. In Europe, no study exists which show PN or BC emission trends of the general light duty vehicle fleet. We provide the capability to do that und we show first results.

3) We presented a measurement system which is capable of determining emissions of
   - different vehicle types (light duty vehicles, heavy duty vehicles, …);
   - low-emission vehicles that meet the latest European emission standards;
   - vehicles in rather dense traffic situations with a distance between the vehicles down to 3 s.

   Up to now, literature PS studies mainly measured HDVs. The used setups restricted measurements to rather large distances (> 7-10 s) between the vehicles because of the slow response time of the used instruments. This limited the application to certain types of vehicles or low traffic areas.

In addition, the setup enables automatic post-processing of the emission data. Commonly in PS studies only a camera was used for vehicle identification. These are often not able to capture all vehicles in dense traffic because they are too slow. Automatic post-processing of the emissions of individual vehicles is not possible if the emissions cannot be attributed to an individual vehicle. We use light barriers for exact pass time determination in addition to an ANPR camera.

Based on the reviewer comments, we have decided to publish the software framework. A first version of the software framework is published here: https://gitlab.com/tug-ems/point-sampling.git. The software framework is being further developed and additional functions will be added as soon as they are ready.

The main adaptions to the manuscript are:

- We have rewritten section "*2.2.2 Emission event processing*" based on the reviewer comments including a more detailed flow chart. This provides a detailed description of the peak detection algorithm.
- We have moved Appendix C to the results section as a new section "*3.1 TUG-PDA emission separation capabilities*". This also includes an assessment of the influence of plume superposition on the results (Figure 6).
- We have merged the (old) sections "*3.2.1 Sampling position*" and "*3.2.2 Measurement location*" into a more compact section "*3.2.2 Measurement location and sampling position*" and moved less important parts to the Appendix.
- We have updated the results based on a new update of the peak detection algorithm. The results have not changed fundamentally. Based on the review we have improved the algorithm, particularly in the case of BG determination and plume separation. As a result, not separable emissions are better detected, separated or excluded from the results, and fewer negative emission results are caused. Therefore, emission values have shifted upwards. The number of valid measurements decreased.

We applied the following minor adaptions which are not addressed in the comments below:

- General improvement in the use of language.
- We have moved the "*Instrument characteristics*" subsection further up in the Results (from 3.2.4 to 3.2.1). The first sentences, which provide a literature review, have been moved to the Methods section (to *2.1 Measurement setup – Emission measurement - Instrumentation*).
- We have moved the subsection "*3.1 Capture rate*" from the Results section to the Methods section (now "*2.3 Capture rate*").
- We have removed part of the $NO_x$ analysis in section 3.3.2 "Fleet Emission Characteristics" as we feel it does not fit well here.
- We have removed the last two points in the conclusion on BG conditions and misalignment as we think that the key messages should be emphasized at this stage.
* * *
Further exploration of the thousands of measurements made could help to enhance the novelty of this work by highlighting new or potential trends in vehicular emission such as emission control technology deterioration as mentioned in the introduction of this study. Additional findings, revisions and additional review would need to be completed prior to acceptance and publication.

We thank the reviewer for this input. It is also in our interest that further explorations are carried out on the measured data. This paper focuses on the further development and exploration of the PS

method and is not intended to discuss emission trends in detail which we plan to publish in another work in the near future including high emitter identification. We extended in the revised version the methodological section by providing detailed information on the peak detection algorithm including plume separation and background determination. In addition, we publish the software framework so that also others may use it. In this manuscript, we exemplarily show the performance of our system by presenting emission trends for different vehicle types (LDV + HDV) even to the latest Euro emission standards for $NO_x$, BC and PN. We also compared our results with several literature studies. For BC and PN, hardly any literature exists which shows todays real world emissions of light duty vehicles captured by remote emission sensing. We show for the first time remote emission sensing results concerning BC and PN emissions of the European light duty vehicle fleet.
* * *
Additional comments:

Introduction, L 36.

"…by making wrong measurements."

Can this be further explained or cited?

During our measurement campaigns in Europe we had a lot of discussions with different institutions in different countries. We have been told, that in certain countries it is common that during PTI "wrong measurements" – manipulated measurements are performed e.g. the PTI exhaust measurement is performed with another vehicle located next to the vehicle which is tested. This bypasses the PTI measurement of the inspected vehicle.

We adjusted in the revised manuscript: "*…by manipulating measurements.*"
* * *
Methods, 2.1 Measurement Setup, L93.

"Using light barriers restricts the measurement location to single-lane roads or roads with islands between the lanes. Alternatively, vehicle detection can be performed with radar, video, or LiDAR systems."

What does this limitation have on the type of vehicles able to be measured with this system?

Do the other sampling options for vehicle tracking listed have the same capabilities but better capture for more road types. If so, why were they not used? Please sure explain the impacts of this sampling method especially with regards to vehicle population and potential bias.

Light barriers limit the application to single lane roads or roads with islands between the lanes. With other vehicle detection technologies such as Radar or LiDAR our presented approach could be applied to multilane roads. With the simple sampling setup (sample extraction from the side or middle of the road) only the outer lanes could be monitored without structural changes to the road. The application to multilane roads (without islands) has not yet been tested.

There is in general no limitation for specific types of vehicles. The limitation in case of vehicles is the exhaust pipe position. In Europe, most vehicles (passenger cars, trucks, motorcycles) have the exhaust pipe at the bottom of the rear or on the side. The tailpipe points straight back or down. A significant share of the motorcycles have the tailpipe pointing upwards which impedes the measurement (at low heights). This is also described in "3.4.1 Fleet composition and capture rate".

We showed in the application example (Fig. 12-15) that we can measure the most common vehicle types.
* * *
Methods, 2.1 Measurement Setup, L110.

"In general, the closer the sample inlet is to the emission source (tailpipe) the smaller the dilution and the higher the capture rate are."

Also discussed in 3.2.1 Sampling Position and eventually mentioned on L463.

Though true, in the schematic, diagram, and later sections, the sampling inlet is located near the ground, what was the capture rate for vertically oriented tailpipes? How did that influence your sample population? Is there potential for this method to be adjusted to capture all tailpipe orientations?

In Europe, there are practically no vehicles with vertical exhaust pipes, with the exception of non-road mobile machinery. Therefore, we have no statistics for such LDVs or HVDs. We see with our sampling setup a lower capture rate for motorcycles whose exhaust pipes are often sloped upward (and which have a lower exhaust flow rate → described in "3.4.1 Fleet composition and capture rate"). With our sampling approach at the roadside or center of the road, vehicles with vertical tailpipes are not captured. If vehicles with vertical tailpipe should be measured, the sample extraction should be done from the top (e.g. from bridges), as shown in several publications referred to in our manuscript.

We have added in section "3.2.2 Measurement location and sampling position" in Line 427: "*Higher CRs and stronger CO2 signals are achieved at lower sampling inlet heights for most vehicles in Europe. An exception are L-type vehicles (e.g. motorcycles) with tailpipes pointing straight or even upwards.*"
* * *
Methods, 2.2.2 Emission event processing, L187.

"At the same time, care must be taken to ensure that the CO2 plume detected of the passing vehicle is related to the pollutant emission detected. Therefore, checks are implemented which compare the duration of the integrated CO2 and pollutant data and verify if the areas overlap appropriately."

Generally, the procedure of peak identification and peak alignment with passing vehicles needs to be further clarified. Can you explain more on how you know that CO2 has returned to baseline to meet the conditions outlined? Does the end of the pollutant peak only rely on another vehicle pass being detected? If so, what does it mean with regards to truly capturing the extent of a CO2 peak? Other works cited have specific quantitative assumptions for the rise above baseline for pollutants and CO2 as well as for the return to baseline after a vehicle passes, can more quantitative information like this be provided?

We don't necessarily wait until the $CO_2$ concentration has returned to the baseline (background). This is only one of the conditions defined in the algorithm which stops the plume integration. There are three mainly three criteria defined which stop the integration of a plume:

- The concentration level is below the determined BG concentration
- Another vehicle passed and the concentration gradient is again rising
- The maximum defined plume duration is reached. 25 s are used in this study.

In addition, two further criteria are defined which cross-check that the areas of $CO_2$ and pollutant agree (difference between the integrated areas and the stop time of the integration are not allowed to exceed defined values).

The following gradients are set as default values in the software environment and were used in this study. Note that the values in the algorithm are defined for the default time scale used, which is 0.5 s.

- $CO_2$: 8 ppm / s
- BC: 4 ($\mu$g /m³) / s
- PN: 4,000 (#/cm³) / s
- $NO_x$: 12 ppb /s

We have rewritten the section "*2.2.2 Emission event processing*" by providing more detailed and better structured information on the plume detection and separation. Several quality assurance (QA) measures are applied in the software which are described in the revised manuscript:

"

- *Vehicle distance: First, when a new vehicle pass is fetched, it is checked whether the distance to the next vehicle pass is sufficient (≥ 3 s). If this is not the case, the processing of the current vehicle is stopped and the algorithm proceeds to the next vehicle. With this small spacing, there is a large uncertainty that emissions will be attributed to the wrong vehicle due to differences in the sampling delay between vehicles.*
- *Separability: The detected gradient (plume) must not be from a previous vehicle. The processing is skipped if either:*
    - *A rising gradient (start condition) from the previous vehicle is found within a pre-defined time frame (default: 3 s before the vehicle pass of the current vehicle) and the plume directly interferes with the current vehicle.*
    - *Or a significantly higher pollutant concentration was measured in the last period (default: 25 s) than for the current vehicle and the current vehicle is likely to be affected by this emission. This is the case when the emission of the previous vehicle was much higher than the peak of the current vehicle and the BG concentration before the current vehicle is still significantly higher than the BG without vehicles.*
- *Pollutant vs $CO_2$ start time: The pollutant peak must start within a pre-defined window compared to the $CO_2$ peak (default window: -1 to 3 s).*

"

And after the integration:

"

- *Duration: The integration interval must exceed a minimum value (default: 3 s).*
- *Plume strength ($CO_2$ only): The integrated $CO_2$ area must be greater than a defined minimum concentration (default: 80 ppm s).*
- *Pollutant vs $CO_2$: The integration interval of the $CO_2$ emission event including a pre-defined factor (default: $t_{max\,diff}$ =0.6) must not be greater than the integration interval of the pollutant emission event.*
- *Pollutant vs $CO_2$: The $CO_2$ and pollutant integration intervals must overlap to a certain extent (default: by at least 50 percentage)*

"

We also replaced Figure 3 with a detailed flow chart of the peak detection algorithm. We have added Table 2, which gives the default values for emission gradient thresholds.
* * *
Conclusion, L559.

"The core of this software is the TUG-PDA, which determines and separates vehicle emissions down to a distance of 3 s between the vehicles, if appropriate instruments are used."

Does this software have potential to be adapted fit the sampling behaviors of a range of instrumentation? The following bullet point is helpful to understand instrument requirements but if others were to try to adapt or replicate this work, is it fully instrument limited? Thinking back to the example provided comparing the AE33 and the BCK. How could one use the AE33 with this method/software? If this is outside the scope of this work, that is fine but please acknowledge.

The software is not developed for specific instruments. We developed the software for modularity and extensibility, such that new instruments can easily be integrated. Each instrument is defined in a separate Python class. New instruments can be added by copying an existing instrument class and adapting the code for the new instrument. In general, any instrument that provides time series data can be used. Parameters such as "gradient thresholds" (for plume start detection), "minimum time to next vehicle" or the "minimum number of required samples for a valid measurement" must be tuned for instruments with different characteristics (sensitivity, response time).

We provided recommendations for instruments to get the best possible PS results (see Table 1). Instrument characteristics such as response time (a large response time requires a larger distance between the vehicles) and sensitivity (a low sensitivity makes it difficult to quantify emissions of vehicles with low emissions / new emission standards) have a great influence on the results. Parameters of the software framework can be adjusted for different instruments. The AE33 can be used with this software framework for locations with low traffic density or for sampling specific vehicles (e.g. trucks with vertical tailpipe when sampling from the top). The rather large response time (~ 7 s) of the AE33 restricts the application to traffic situations with a distance between the vehicles of more than 7 – 10 s.

We added in section "2.2 Data analysis": "*The software has not been developed for specific instruments and in general any measurement device that provides continuous measurement data can be integrated. However, we strongly recommend to consider the recommendations in Table 1 when selecting instruments to achieve the best possible results.*"

We added in section "3.3.3 Instrument characteristics": "*However, the Aethalometer can also be used for PS, as has already been demonstrated in several studies. The traffic density must be low enough (distance between vehicles greater than 7-10 s) or only certain types of vehicles (e.g. HDVs with vertical exhaust pipes) are measured, which naturally entail a greater distance between exhaust plumes.*"

**Referee 2**

General Comments:

This manuscript describes a sampling system that can be used to capture and evaluate in-use emissions by thousands of vehicles. The authors include detailed data that quantify the impact of important sampling location and configuration features and environmental conditions on the success rate of their point sampling method, and a first look at the emission trends that have been captured by this system. Such an automated platform for capturing on-road vehicle emissions and determining emission factors would be extremely useful for both regulators and researchers for tracking fleet trends and identifying high emitters.

The key innovation presented in this paper is their automated peak detection algorithm. However, not enough information is given for a reader to replicate this methodology independently, and this reviewer also has several questions about how the algorithm functions (see below). If the algorithm will be made publicly available and if the below questions/comments are addressed, then I believe that the manuscript could pair well with it. But if the algorithm will not enter the public domain, then I think the manuscript requires major revision to be a more complete methods paper that could be independently duplicated by others. Alternatively, if the authors do not intend for this paper to be a methodology paper, then less focus should be spent on the results for their sampling platform/algorithm performance and more should be spent on the vehicle emissions that were sampled. They have a rich dataset for tens of thousands of in-use vehicles at multiple locations in Europe, which could easily be the focus of the paper. The manuscript in its current form seems split between the two narratives and incomplete for both, and it would be more compelling and impactful to focus on either the method or the fleet results.

We thank the reviewer for this very detailed review, comments and suggestions. This review is very helpful to improve the manuscript. As suggested, we have made the software framework public available as we also believe that this can help researchers and institutions to further develop emission monitoring concepts. In addition we described the peak detection algorithm in more detail in the methods section of the manuscript. We answered the comments below and we revised the manuscript accordingly.

Based on the reviewer comments, we have decided to publish the software framework. A first version of the software framework is published here: https://gitlab.com/tug-ems/point-sampling.git. The software framework is being further developed and additional functions will be added as soon as they are ready.

The main adaptions to the manuscript are:

- We have rewritten section "*2.2.2 Emission event processing*" based on the reviewer comments including a more detailed flow chart. This provides a detailed description of the peak detection algorithm.
- We have moved Appendix C to the results section as a new section "*3.1 TUG-PDA emission separation capabilities*". This also includes an assessment of the influence of plume superposition on the results (Figure 6).
- We have merged the (old) sections "*3.2.1 Sampling position*" and "*3.2.2 Measurement location*" into a more compact section "*3.2.2 Measurement location and sampling position*" and moved less important parts to the Appendix.

- We have updated the results based on a new update of the peak detection algorithm. The results have not changed fundamentally. Based on the review we have improved the algorithm, particularly in the case of BG determination and plume separation. As a result, not separable emissions are better detected, separated or excluded from the results, and fewer negative emission results are caused. Therefore, emission values have shifted upwards. The number of valid measurements decreased.

We applied the following minor adaptions which are not addressed in the comments below:

- General improvement in the use of language.
- We have moved the "*Instrument characteristics*" subsection further up in the Results (from 3.2.4 to 3.2.1). The first sentences, which provide a literature review, have been moved to the Methods section (to *2.1 Measurement setup – Emission measurement - Instrumentation*).
- We have moved the subsection "*3.1 Capture rate*" from the Results section to the Methods section (now "*2.3 Capture rate*").
- We have removed part of the NO$_x$ analysis in section 3.3.2 "Fleet Emission Characteristics" as we feel it does not fit well here.
- We have removed the last two points in the conclusion on BG conditions and misalignment as we think that the key messages should be emphasized at this stage.
* * *
Additional general comments:

Many acronyms are introduced but only used a couple of times (e.g., PC, PTI). This can be confusing for the reader, so this reviewer suggests only using those acronyms that are frequently used (e.g., HDVs, PS, etc.) and minimizing the introduction of others.

We thank the reviewer for this input.

We have reduced the usage of several acronyms in the revised manuscript such as PC, PTI, CPC, LoD.
* * *
I did not think enough information was given about many important details for the algorithm and have many specific questions that are listed below. Overall, I have questions about:

- How the background concentrations are determined and applied
- Peak separation and overlap
- QA/QC steps and what is considered successful versus what is screened out of the analysis presented in Section 3.3

We thank the reviewer for this input and we will provide more details in the revised manuscript regarding these open questions. These questions are addressed by several comments and answers below.

In the revised manuscript we provide detailed information on the above mentioned points in section "*2.2.2 Emission event processing*" and section "*3.1 TUG-PDA emission separation capabilities*".
* * *
This manuscript is well written, but is long and could be significantly shortened. In its current form, it's difficult for the reader to pull out the key results and insights. These are well summarized in the Conclusions, but they're otherwise not obvious with the current density of results, discussion, and

figures. In many cases, results could be summarized more concisely with simpler statements like "results were comparable across all conditions" and the supporting figures could be moved to the Appendix, rather than describing them each in detail. In other cases, results and figures could be combined and presented together, rather than discussed in detail separately. For example, the results and discussion about sampling position and measurement location could be combined and streamlined to more directly and efficiently conclude that the capture rate is higher when you sample closer to the emission source.

We agree and we thank the reviewer for these suggestions. We agree that especially the interpretation and discussion of the measurement location and sampling position is too long and can be written more concisely.

We merged (old) sections "*3.2.1 Sampling position*" and "*3.2.2 Measurement location*" to "*3.2.2 Measurement location and sampling position*". We described the results in a more compact form. We have moved (old) Figures 5a (Distribution of $CO_2$ concentrations of the sampling positions) and 9 (Background concentrations) to the Appendix (see comments below).
* * *
Specific Comments:

Introduction, Lines 20–24: Emission control system performance decline via tampering or malfunction is emphasized as the only source of high emissions of NOx and PM, whereas these high emissions can also simply come from older engines without these newer after-treatment controls (e.g., non-DPF-equipped vehicles). In other words, skewed fleet emissions and the high emitter problem are not solely due to degrading DPFs or SCR systems that have been tampered with or are failing.

We agree and we thank the reviewer for this input. A significant share of emissions come from old vehicles.

We adjusted the description in the introduction to: "*Nitrogen oxide (NOx) emissions remain a widespread problem, especially for diesel-powered vehicles, where tampered, defective and old vehicles are the main source of high emission levels (Meyer et al., 2023). For PM it is well known from literature, that a small share of vehicles (< 20 %) contribute to the vast amount (60-90 %) of emissions (Park et al., 2011; Burtscher et al., 2019; Boveroux et al., 2019; Bainschab et al., 2020). This is due to malfunctioning after-treatment systems, such as defective diesel particulate filters (DPF) and old vehicles with degenerated or outdated technologies.*"
* * *
Introduction, Lines 28 and 33: Is the interest really in PN concentrations (which vary with dilution) or emission rates?

We agree that you cannot purely look at concentrations. In remote emission sensing it makes no sense without reference quantity (e.g. $CO_2$). But often concentrations are of interest or are even used for regulations. In environmental sensing concentration thresholds are used (e.g. $PM_{2.5}$, $PM_{10}$, $NO_x$) by authorities (e.g. EU) for monitoring the current situation and sanctioning member states. During the periodical technical inspections of vehicles, exhaust measurements are performed directly at the tailpipe to check whether the vehicle complies with the regulations. Therefore, for example in Germany for Euro 6 vehicles a PN threshold of 250.000 particles/cm³ is defined.

We adapted Line 28 to: "*Particle number (PN) and black carbon (BC) are two PM metrics of particular interest.*"

Methods, Figure 1 and Lines 108–109: How do you functionally position the sample inlet in the middle of the road where cars are driving? I'm assuming you put some sort of rigid protector over the tubing that vehicles drive over. But does that alter how they are operating (e.g., slow down while passing by, etc.), or is it small/inconspicuous enough that vehicles do not "see" it and do not change their driving patterns.

The sampling tube was put into a small cable duct (height: 2 cm, width: 13 cm) which was then fixed with duct tape onto the road.

We did not observe a significant influence on the driving behaviour with our setup and with this sample extraction from the road centre. We did observe an influence on the driving behaviour when people were standing outside or a camera was located next to the road (our camera was in the drivers cabin).

In section "*2.1 Measurement setup*" – "*Emission measurement*" – "*Sampling*" we added: "*When sampling from the center of the road, we covered the tube with a small cable duct that was taped to the road.*"

In section "*3.3.1 Measurement location*" we added: "*We found no significant influence on the driving behavior when the sampling was done from the center of the road through the covered tube.*"
* * *
Methods, Table 1: For vehicles without SCR or inactive SCR systems, $NO_x$ concentrations can be even more elevated, up to ~10 ppm.

We thank the reviewer for this note. Indeed, there are individual vehicles (< 10) where we measured concentrations greater than 5 ppm (including dilution).

We have changed the recommended range for $NO_x$ in Table 1 to 10 ppm.
* * *
Why is the algorithm called TUG-PDA? PDA is defined as peak detection algorithm, but TUG is not defined.

As „peak detection algorithm" is very generic we wanted to have a specific name for the algorithm and added TUG which stands for "Technische Universität Graz" (german name of the university). Therefore we wrote in the abstract "our" PDA.

On page 3, we have added a footnote explaining the origin of "TUG".
* * *
Is TUG-PDA deployed in real-time, or is it used after data is collected? In other words, are peaks being detected and integrated live, or is this used as a post-processing step after data has been collected?

The algorithm is applied during post-processing. But could be in principle be applied in (near) real time with a delay of some seconds because of sampling and instrument delay.

Methods, Lines 180–181: If you smooth the data, does that not affect the peak area for the emission factor calculation?

Yes, it slightly affects the results but single measurement failures (outliers) from instruments are smoothed and the PDA algorithm is affected by short dips or peaks in concentration data. Currently, the data is smoothed with a rolling gauss filter for 5 samples (2.5 s). If this is done over a larger sample set (e.g. to match slower with faster instruments), the influence will increase. It also equates the differences in response times between the instruments. It also compensates for differences in response time between instruments.

Methods, Lines 195–207: I found the discussion and figures in Appendix C to be very helpful for better understanding how TUG-PDA operates, and suggest moving them (or a streamlined version of them) up to the main manuscript.

We thank the reviewer for this input.

We moved Appendix C into the results section to "*3.1 TUG-PDA emission separation capabilities*". We have rewritten the section for better understanding.

A 3-second delay seems really small between vehicles, when most peak events occur over 5–10 seconds. The only concern listed was misattribution of the captured pollutant peaks to the incorrect vehicle, rather than overlapping peak events. You detail peak separation in Appendix C, but not enough information is given here. How can your algorithm distinguish between vehicles if the $CO_2$ and pollutant concentrations do not return to background before starting a new peak integration? It's unclear to me how you can get accurate integrations when peaks overlap with vehicles passing in rapid succession. Even if you assume a background concentration, the tails of the peak itself will be cut off. How are you certain that you have an accurate emission ratio under these circumstances?

We thank the reviewer for these questions. It is true that the emission calculations get more inaccurate with overlapping plumes. But we found that statistics are not significantly affected by overlapping plumes if the parameters of the algorithm are deliberately set such that the plumes can be separated properly (in accordance with e.g. instrument response times). How the PDA algorithm is configured is also a trade-off between statistics (number of samples) and increasing inaccuracies. We are sure that improvements on the plume separation can be made which should be addressed in the future. The plume separation can be adapted in the framework with several parameters such as:

- "Minimum time to next vehicle" → Sets the minimum distance between vehicles up to which the PDA attempts to resolve the emissions.
- "Minimum number of required samples for valid measurement" → Sets the minimum number of samples required for a measurement to be valid.
- "Time delta for previous vehicle check" → Defines the time up to which the PDA "looks back" to the previous vehicle and checks if the emission could interfere with the current vehicle.

This can be very useful to adapt the post-processing for locations with dense traffic or low traffic to get sufficient number of samples. The plume separation capability of the algorithm enables the application to sampling locations with also higher traffic volumes.

We have extended section "*3.1 TUG-PDA emission separation capabilities*" with a comparison of results of measurements with and without interference (Figure 6). In section 3.1, only the

interferences for $CO_2$ plumes are considered. Appendix C also deals with interferences of pollutants (BC). Therefore, we added a feature to the algorithm that marks whether or not the plumes overlap.

In the software default parameters are defined which are also stated in the revised manuscript.
* * *
Have you characterized how different fractions of peaks captured and assumed baseline concentrations impact resulting emission factors, using the subset of peak events that were 100% isolated with all pollutants starting and ending at the background condition?

We thank the reviewer for this suggestion. We have not evaluated that in detail, but that is a very interesting suggestion for future investigations.
* * *
It's unclear to me how TUG-PDA defines the start and stop time of peak events. Do the $CO_2$ start/stop times define the peak event, and then those are mapped onto each pollutant time series with the previously determined time adjustments due to different instrument responses? Or are the $CO_2$ and pollutant peaks handled independently by the algorithm, with $CO_2$ first for successful plume capture and then each pollutant if the $CO_2$ peak analysis was successful?

Start time: The vehicle pass time from the light barriers is used as a starting point. The PDA searches around the vehicle pass time for a sequence of increasing gradients (2 rising gradient above threshold or a very large gradient (> 10 x threshold)). The PDA "start range" time period was set in this study to -1 s to 6 s around the vehicle pass (see Figure 4). The first rising gradient is used as start time for integration.

Stop time: There are three mainly three criteria defined which stop the integration of a plume:

- The concentration level is below the determined BG concentration
- Another vehicle passed and the concentration gradient is again rising
- The maximum defined plume duration is reached. 25 s are used in this study.

In addition, two further criteria are defined which cross-check that the areas of $CO_2$ and pollutant agree (difference between the integrated areas and the stop time of the integration are not allowed to exceed defined values).

$CO_2$ emissions are first processed. Pollutant emissions are only processed for vehicles with valid $CO_2$ plume. After the $CO_2$ processing the pollutant emissions are processed. The start and stop times for the pollutants are independently determined compared to $CO_2$. This is verified by QA measures. For the processing of the pollutant emissions there are two separate cases:

1) **Significant emitter (pollutant peaks detected):** For distinct pollutant plumes which were identified by the PDA: Integration of the pollutant concentration independent of the $CO_2$ signal. Afterwards the time frames of $CO_2$ and pollutant are compared and if start /stop time are in range and the duration is ok then the emission ratio is valid. After the integration of the pollutant length, start and stop times of $CO_2$ and pollutant are compared. There are maximum delay times defined and it is not allowed that the pollutant area is longer than the $CO_2$ area. These times can all be adjusted with several parameters in the algorithm.
2) **Low emitter:** For vehicle passes where only a $CO_2$ peak was captured (and no pollutant peak): Pollutant emissions are integrated over the same time period as the $CO_2$ signal.

We have rewritten section "*2.2.2 Emission event processing*" by providing detailed and better structured information on the peak detection algorithm. We have replaced Figure 3 (flow chart of the PDA) with a new flow chart that describes the procedure in more detail.
* * *
A background concentration based on the minimum value before the passing time is likely biased low, which would overestimate the true integrated area of each pollutant. For instance, background BC concentrations might bounce around –2 to 4 µg m$^{-3}$ on a secondly basis. A running average shortly before the start of the peak would more accurately capture the true baseline ~0–1 µg m$^{-3}$, rather than assuming a value of –2 µg m$^{-3}$, if that was the minimum concentration before the passing time. Similarly, for $CO_2$ concentrations under high traffic conditions, the background concentrations can vary by ± 50 ppm. The choice in background value can have large impacts on the resulting emission factor, and these questions need to be better addressed in the manuscript, especially when considering the limit of detection for this system when calculating near-zero emission rates with low emission events (i.e., good DPF and SCR performance) for those plumes with weak capture (i.e., small $CO_2$ peak area) events.

We agree that the minimum value is likely biased low and causes overestimated integrated areas. We also agree that the BG determination has a large impact on the results, especially for low emitter. We adapted the algorithm to use the minimum of a running average before the plume based on comparison of results with test vehicles equipped with PEMS. The background determination is one area (besides plume separation) where further improvements are possible. This should be evaluated in more detail in the future (especially for overlapping plumes).

We provide more detailed information on the BG determination in section "*2.2.2 Emission event processing*" and in section "*3.1 TUG-PDA emission separation capabilities*" for overlapping plumes (see comments blow).

We also added a suggestion in the conclusion regarding improvements: "*Further development of the TUG-PDA could address improvements in the determination of BG for overlapping plumes (e.g. using a linear approximation between the start and end of the peak) or the use of adaptive thresholds and parameters depending on the measurement location.*"
* * *
In Figure 4, you plot only positive values for BC, even though it looks like concentrations dip below zero for the background values before the peak events. Is this just a formatting choice for plotting this example, or does this mean that your algorithm ignores negative values? In this example, it doesn't seem to matter for the peak integration, since there is a strong BC signal. But for the case where there is no BC peak (or other pollutant) that corresponds to a $CO_2$ peak, those near-zero concentrations can be positive or negative. The negative values are valid and should be included in an emission factor calculation.

We thank the reviewer for this input and comment. The time series data are taken as measured by the instruments besides smoothing / interpolation which depends on the sampling frequency, response function, … . Therefore, negative values are also included in the calculations. Important is always the difference between background and measured plume and not the absolute concentrations. The example is a bit misleading as the background is not subtracted in the figure. In addition, the black carbon tracker is currently slightly high biased (1-2 µg/m³). But the linearity is very good (basically 1 to reference equipment). Therefore, for point sampling where only BG subtracted concentrations matter it should not matter at all.

How were the thresholds for positive concentration gradient (Line 199) and minimum $CO_2$ integrated peak area (Line 214) determined? Are these also dependent on sampling configuration, driving and/or engine load conditions, environmental conditions, etc.?

The thresholds were determined based on tuning of the parameters and manual evaluations of the PDA by reviewing many (100s) of vehicle passes. Currently, the same thresholds were used for all sampling configurations etc. But this is a very good point and can be tested and adjusted in the future.

The following gradients are set as default values in the software environment and were used in this study:

- $CO_2$: 8 ppm / s
- BC: 4 (µg /m³) / s
- PN: 4,000 (#/cm³) / s
- $NO_x$: 12 ppb /s

We added Table 2 with the used thresholds in section "*2.2.2 Emission event processing*" and added: "*The thresholds were determined based on a large number of manual reviews of TUG-PDA results.*"

We refer to possible adaptive thresholds in the conclusion: "*Further development of the TUG-PDA could address improvements in the determination of BG for overlapping plumes (e.g. using a linear approximation between the start and end of the peak) or the use of adaptive thresholds and parameters depending on the measurement location.*"

Line 216–217: If instruments have different response times, the pollutant peak could extend beyond the $CO_2$ If you've smoothed the data (Lines 180–181) to force this scenario to not happen, how have you verified that this does not affect the corresponding emission ratio? You extensively discuss time alignment in Appendix E, but not in terms of this question.

We used instruments with similar response times (0.9 s to 2 s). By smoothing the data, we adjust the instrument responses. Sampling delays and response times of the instruments are aligned in the pre-processing steps of the software. In addition, $CO_2$ and pollutant data are separately processed. For overlapping plumes, results may deviate despite all these actions. If instruments are used with significantly different response times (difference > 2-3 s), the responses must be matched accordingly.

We added in section "*2.2.1 Pre-processing*": "*If instruments with large differences in response times (Δt > 2 s) are used, the response function of the instruments must be aligned.*"

Lines 271–272: Do you determine if plumes can be separated and assigned clearly to a specific vehicle algorithmically via rules in TUG-PDA or with visual/manual inspection of TUG-PDA results? How do you QA/QC the data to verify that only valid emission factors that can be fully attributed to individual vehicles are included in the final dataset?

We thank the reviewer for pointing out the missing information regarding QA measures.

The whole emission processing is done automatically. Manual inspections are only applied for verification checks. There are several checks and QA measures included:

- Emissions of one vehicle are only considered if the spacing to the previous vehicle is larger than a predefined value (in this study 3 s).
- A rising plume (two positive gradients above threshold in the given sequence or a very strong gradient (> 10 times the threshold)) of must be in a defined window of the vehicle pass. For this study -2 s to 3 s around the vehicle pass are used.
- The detected gradient (plume) must not be from a previous vehicle. This is checked by checking if a vehicle with less than 10 s distance caused a plume which was not considered yet.
- The captured plume must have a minimum length of 6 samples (3 s at 2 Hz measurement rate)
- For pollutant:
    - If a pollutant plume was detected. The length of the plume must be smaller or equal to the $CO_2$ plume length.
    - The pollutant plume must start within a pre-defined window compare to the $CO_2$ plume
    - They must at least overlap by 50 percent
    - The pollutant plume lasts not longer than a predefined value (3 s in this study) compared to the $CO_2$ plume
- For translation of emission ratios to emission factors: ANPR camera data and light barrier data are separately captured. The ANPR camera pictures are related to the light barrier pass times. There is a time difference between the ANPR camera time of the vehicle and the light barrier pass time. The light barrier pass time is very exact and one very important part that automated post-processing is feasible. The time of the ANPR camera capture is varying. Our algorithm relates the ANPR pass time to the LB time with the usage of the measured vehicle speed and acceleration by the light barriers. A matching is performed between these two times and the pair (between ANPR and LB) with the smallest time difference is matched together.

All of these criteria were verified by manually reviewing 100s of PDA results and adapting the algorithm accordingly.

We added in the revised section "*2.2.2 Emission event processing*" a description of the QA measures during the start condition (peak search):

"

- ***Vehicle distance:*** *First, when a new vehicle pass is fetched, it is checked whether the distance to the next vehicle pass is sufficient (≥ 3 s). If this is not the case, the processing of the current vehicle is stopped and the algorithm proceeds to the next vehicle. With this small spacing, there is a large uncertainty that emissions will be attributed to the wrong vehicle due to differences in the sampling delay between vehicles.*
- ***Separability:*** *The detected gradient (plume) must not be from a previous vehicle. The processing is skipped if either:*
    - *A rising gradient (start condition) from the previous vehicle is found within a pre-defined time frame (default: 3 s before the vehicle pass of the current vehicle) and the plume directly interferes with the current vehicle.*
    - *Or a significantly higher pollutant concentration was measured in the last period (default: 25 s) than for the current vehicle and the current vehicle is likely to be affected by this emission. This is the case when the emission of the previous vehicle*

> *was much higher than the peak of the current vehicle and the BG concentration
> before the current vehicle is still significantly higher than the BG without vehicles.*
> - ***Pollutant vs CO₂ start time:*** *The pollutant peak must start within a pre-defined window
>   compared to the CO₂ peak (default window: -1 to 3 s).*

*"*

And after the integration:

*"*

> - ***Duration****: The integration interval must exceed a minimum value (default: 3 s).*
> - ***Plume strength (CO₂ only)****: The integrated CO₂ area must be greater than a defined minimum
>   concentration (default: 80 ppm s).*
> - ***Pollutant vs CO₂****: The integration interval of the CO₂ emission event including a pre-defined
>   factor (default: $t_{max\ diff}$ =0.6) must not be greater than the integration interval of the pollutant
>   emission event.*
> - ***Pollutant vs CO₂****: The CO₂ and pollutant integration intervals must overlap to a certain extent
>   (default: by at least 50 percentage)*

*"*
* * *
Results: I suggest combining and streamlining sections 3.2.1 and 3.2.2, as the results and discussion are presented together, can be a little difficult to tease apart as they are currently discussed, and the existing text can be a little repetitive. I think choice of measurement location as described in Lines 314–320 is probably the most important factor in terms of successful point sampling, and it is best to describe those characteristics first. The sampling position details at a given location are more nuanced, and could be combined to better complement each other after establishing what a good sampling location requires in terms of road properties, traffic conditions, and vehicle operation. For instance, the discussion on Lines 354–361 about road width are very similar to the discussion in Section 3.2.1 about tailpipe and sampling direction and sampling heights.

We thank the reviewer for this input and we agree that this can be simplified. With this detailed description we wanted to emphasize that there are several factors which influence the quality of the measurements. In addition to the measurement location, the sampling position is equally important. This is for example shown in Figure 6 for measurement location 3: When sampling from the road center, the capture rate is three times as high as compared from the right side of the road (especially at large lane widths).

We merged and shortened (old) Sections "3.2.1 Sampling position" and "3.2.2 Measurement location" to "3.2.2 Measurement location and sampling position". We have moved the figures on changing background conditions and sampling signal strength to the Appendix (now Figure D1 and E1).
* * *
Results: Consider combining Figures 6, 7, and 8b to be side-by-side, since the three are very similar and discussed together in Lines 305–312. It's difficult to synthesize all of the information presented in the current form while flipping back and forth between pages and plots.

We thank the reviewer for this suggestion and we agree that putting these figures side-by-side can simplify the interpretation.

We put the figures about sampling position, sampling height and lane width side-by-side. We also put the figures about VSP and median vehicle distance side-by-side. We moved (old) Figures 5a (Distribution of $CO_2$ concentrations of the sampling positions) and 9 (Background concentrations) into the Appendix.
* * *
Results, Figure 6: The trend line seems like it might be showing the combined influence of sampling position and height. In particular, all of the middle sampling points occur at near-ground sample heights with high capture rates, compared to the left and right sampling results that span higher sampling heights and a broad range of capture rates. The combination of sampling position and height might be confounding this result/trend. What would these results look like if the left and right sampling configurations were also conducted at heights < 1cm, like the middle sampling results? How does the trend line shift if the middle results are excluded and only the >4 cm samples are included?

We agree with the assumption that the figure shows a combined influence of position and height. The influence of height can be seen particularly for measurement locations 1 and 2. The sampling location was slightly shifted up the street and the sample extraction was done at lower heights (described in lines 310 – 312). Roadside sampling positions were not conducted at heights < 1 cm mainly because of 2 reasons. 1) Often it is not possible due to the road conditions (e.g. side walk, …) to sample at these low heights. 2) To be able to measure independent of rain, measurements were not performed at these low heights.

We added a second trend line in Figure 8.
* * *
Results, Lines 325–326: If measurements are often made after a crossroad or traffic light, could there be a bias in the emission profiles observed? How does that driving mode compare to "typical" operation?

We agree with this assumption of the reviewer. This is a general problem of RES that the emission trends could be (high) biased by the sampling locations. In RES, measurements must be conducted such that the vehicles are under certain load to capture $CO_2$ emissions.
* * *
Results, Lines 382–383: Is the difference in capture rate noted for dry vs rainy conditions statistically significant? If not, I would suggest a slight re-wording of this paragraph that instead emphasizes that all of the results are comparable (like you do with the $CO_2$ and BC results), rather than pointing out minor differences in capture rate. Also, in this paragraph, you describe differences in average values for capture rate and $CO_2$ concentration, but median differences for BC emission ratios. Is there a reason for not reporting mean differences in BC emission ratios?

Results, Figure 10b: Can you clarify what is meant by the y-axis label of "mean BC ratio"? I assumed that these are distributions of measured emission ratios from individual peak events, but please describe what has been averaged if they are instead distributions of mean ratios.

We thank the reviewer for these suggestions. No there is no reason for not reporting mean differences. The description of Figure 10b) lacks the information that the result was achieved with a Monte Carlo simulation. An equal number of samples was 1,000 times drawn from each measurement location and the mean EF was calculated from these subsets. The boxplots show the

distribution of these calculated mean EFs. In this way, all measurement sites contributed equally to the results, regardless of population size.

As suggested, we changed the description to: "*We found that the PS measurements were not significantly influenced by rain. CRs are comparable during rainy (29.3 %) and dry (29.8 %) conditions (Fig. 10, left).*"

We added as description for Figure 10: "*We were particularly interested in discovering whether these conditions impacted PM emissions. For this purpose, we performed a Monte Carlo simulation by drawing 100 samples of the measured ERs of the passing vehicles from the different measurement sites 1,000 times. We calculated the mean ERs from the 100 samples and the distribution of the mean values is shown in Fig. 10 in the right plot. Statistically, no significant difference was observed between the ERs calculated in dry and rainy conditions with median values of 110 and 134 mg (kg CO$_2$)-1, respectively.*"
* * *
Results, Line 477–478: If the emissions from previously passing vehicles are interfering with the measurement of the current vehicle, shouldn't your algorithm and QA/QC process screen those results out as an unsuccessful capture? If there is interference from other vehicles, then you do not have an accurate measure of an individual plume that can be attributed to the target vehicle and it should not be included in your results. Or, you can consider fleet trend results from combined plumes like in Dallmann et al (2011), but not attribute any vehicle-specific information from license plate data to those emission factors.

This is an interesting aspect. We explained the plume separation parameters and criteria in previous comments. The software allows to define several parameters to tune the plume separation. As mentioned in previous replies, this is a trade off between accuracy and number of measurements. We omitted plumes which are not separable or which overlap "too strongly". We use overlapping plumes where distinct peaks can be found for the individual vehicles and which meet our plume separation criteria ("Minimum time to next vehicle", "Minimum number of required samples for valid measurement", "Time delta for previous vehicle check").

The error coming along with overlapping plumes may cause a deviation for individual vehicles, but on a statistical basis for the whole fleet it has a relatively small influence (see Figure 6).

In addition, it should be kept in mind that interferences from other vehicles are also present in results from (commercial) open-path remote emission sensing devices.

As described above, we have improved the PDA for plume separation. We described in detail the plume separation criteria in section "*2.2.2 Emission event processing*" and showed the influence on the results in section "*3.1 TUG-PDA emission separation capabilities*".
* * *
Appendix A, Line 620: "When plumes overlap or impacts from other sources occur, this concentration may be underestimated." This is an important point that I think should be emphasized in the main manuscript when describing how your algorithm handles vehicles that pass by in rapid succession, especially if you do not exclude them from your results.

We thank the reviewer for this suggestion.

We have rewritten section "*2.2.2 Emission event processing*" and moved a revised version of "Appendix C" into the Results section to "*3.1 TUG-PDA emission separation capabilities*". This should provide detailed information on how overlapping plumes are handled.
* * *
Appendix C, Figure C1:

I'm confused by the shaded areas for CO2 that extend all the way down to ~400 ppm CO2. From the time series, the background concentration looks to be more ~450 ppm, depending on the passing vehicle. Is your algorithm assuming a background concentration of ~400 ppm for all four vehicles? If so, this is an overstatement of the CO2 peak areas. If not, and this is just a figure formatting choice, then I suggest instead either: (1) plotting background subtracted concentrations, (2) adjusting the secondary y-axis range so that the plot looks more like Figure 4, or (3) cutting off the shaded blue areas to only include the above-background portions of the peaks to represent the true peak areas included in the emission ratio calculations.

We thank the reviewer for these suggestions. In the highlighted areas the background was not subtracted. In Figure C1 a), the determined background for the vehicles varies and is between ~435 and 445 ppm.

In the revised manuscript, we have subtracted the determined BG in the adjusted plot (now Figure 5).
* * *
Does the weak capture for V2 pass the minimum requirements for calculating an emission factor? That rise in $CO_2$ (~10 ppm) does not look strong enough above normal noise in background concentrations to be a successful capture. Would this be flagged in QA/QC?

We thank the reviewer for pointing this out. This is a border case and it depends on the configuration of the PDA if emissions from vehicle V2 are considered valid. In the presented results in the manuscript there was a bug in the background determination and therefore vehicle V2 was considered valid. In the latest updated version of the PDA the bug was fixed and we applied stricter rules (higher thresholds for $CO_2$ and pollutants, higher threshold for the $CO_2$ area) for omitting such cases.

In the revised manuscript results are updated including (old) Figure C1 a ($\rightarrow$ now Figure 5 a). Emissions from vehicle V2 are not considered valid.
* * *
A4 is not shaded in or labeled, as noted in Lines 668–669.

We thank the reviewer for pointing this out.

In the revised manuscript emissions from vehicle V4 are also highlighted.
* * *
Technical Corrections:

This might be a journal formatting requirement/preference, but it is sometimes hard to discern paragraph breaks without extra spaces between paragraphs or an indent at the start of a paragraph.

We have used the template from the journal.

We tried to improve the structure of the text by restructuring the paragraphs.

Figure placement throughout the manuscript doesn't always make sense. For instance, Figure 12 is discussed in Section 3.2.3 but appears halfway through Section 3.2.1, while Figure 13 is discussed in Section 3.2.3 but appears on the next page in the middle of Section 3.3. I realize this may be a journal formatting issue rather than one that the authors can address, but wanted to point it out in case it could be adjusted.

The figure placement is for the manuscript mainly our fault/choice.

We tried to improve figure placement in the revised manuscript.

Suggest replacing occurrences where "1000s" was used with the word "thousands" (e.g., Abstract Line 9 and Intro Line 75)

We replaced occurrences of "*1000s*" with the word "*thousands*".

Abstract, Line 13: define NOx as "nitrogen oxides (NOx)"

We replaced "*$NO_x$*" with "*nitrogen oxides ($NO_x$)*".

Introduction, Line 19: define NOx as "nitrogen oxides (NOx)"

We replaced "*$NO_x$*" with "*nitrogen oxides ($NO_x$)*".

Introduction, Line 23: suggest "malfunctioning"

We have replaced "*malfunction*" with "*malfunctioning*".

Introduction, Line 44: define emission factors as "(EFs)" since that is how it is used throughout the paper

We introduced in Line 45 "*(EFs)*".

Results, Line 333: revise to "6,500" or "6500" depending on number format used throughout the manuscript; note that there are some inconsistencies throughout the manuscript that should be made uniform (e.g., "3000" on Line 334 versus "100,000" on Line 275)

We have changed the numbers above one thousand to the number format "6,500".

Line 480: should be "…150 mg (kg fuel)-1"

This error has been corrected.

Line 587: consider word choice substitution for harsh, maybe something like "challenging"

We have removed this paragraph.

---

## Referee Report (RR1)

This study developed and demonstrated a point sampling method to automatically measure emissions from a large-scale of individual vehicles. In this works, the authors present their system that can be used for particulate matter (PM) and gas emissions measurements, which is notably independent of vehicle type. They find that when using their peak detection algorithm (TUG-PDA), they can separate vehicle-specific emissions down to a spacing of just a few seconds between vehicles. In this study, they present initial findings from the use of this method that collected ~100,000 vehicle records from several measurement locations, mainly in urban areas. When compared to equivalent remote sensing measurements, the authors found good agreement even with the newest standards which are harder to capture due to their lower emissions and the current remote sensing abilities. This paper is well written and organized.

This manuscript presents novel work on the development of a plume detection system. The authors have done a lot of work to respond to and update their work based on the last round of revisions. With that being said, if the authors are able to update their work with the minor revisions listed in this report, this manuscript should be accepted for publication.

Line 19:
Define NOx here instead of in line 21

Line 47-48: "Other  PM metrics  such  as PN or BC cannot be accurately determined using these systems..."
This point needs to be further clarified for the reader.

Table 1 /Table 2: Because Table 2 is not referenced in the text and is just hanging in the section that is placed in, it's recommended to add to Table 1 or putting it in the Appendix and referring to it in the main text.

2.2.1 Pre-processing:
This section has a lot of technical information that does not fit well into the bulk of the manuscript. It's recommended to simplify to the following:
1. Raw data from the instruments are time aligned
2. The CO2 is the default time resolution (keep statement about if instruments have large response time differences)
3. Outliers are filtered (state metrics for this) and measurements are smoothed

Lines 194-201:
This section leading up to describing the algorithm do not add value to the methods section and ends up being more distracting. The audience should understand the concept of PS by this part of the paper and therefore, there is no need to provide these detailed ideas. It's recommended to cut or slim to one introduction statement on why you developed the algorithm.

Line 211: "There must be either at least two gradients"

Which two gradients are being referred to? As in two pollutants need to be rising and having a gradient? Please clarify in the text.

Figure 3:
This figure can go into the appendix. It is too much to take in and understand in the main text. Also, it's recommended to add step numbers to make it easier to follow the diagram.

Figure 4:
It would be helpful to have the axes go lower than background so the reader can see how the background level is determined and potentially used as a stop (while recognizing that this specific example is not due to background but because of a passing vehicle)

Line 269:
Another recent paper on plume detection was published and should be referenced here. https://www.sciencedirect.com/science/article/pii/S1364815222003000

Section 2.2.3:
This section can be revised to be more direct about exactly what this method does. Also, generally, the methods section still seems quite lengthy. It is recommended to highlight the specific and unique points that applies to this PDA system.

Lines 306-311:
These sentences make the TUG-PDA sounds like it hypothetically can get down to a detection of 3 s but it is not able to in many cases, which contrasts a lot of the high level take aways from this paper. It is suggested to reword in the more active and present tone to express exactly what the system can do and what factors are able to be adjusted for varying situations.

Figure 5a:
For V4, the BC integrated areas seems to be cut off early. Though it may be following the rules, the BC plume both starts later and ends later than the corresponding $CO_2$ plume and therefore, should be integrated to basically be a time shifted version of the $CO_2$ plume instead of being cut off which would lead to an underestimation of the EF (this is discussed later on when the author states "The median EFs were 19% lower than in cases without interference"

Because of this, I believe it is very important for the authors to directly respond to the previous comment from Referee 2: Have you characterized how different fractions of peaks captured and assumed baseline concentrations impact resulting emission factors, using the subset of peak events that were 100% isolated with all pollutants starting and ending at the background condition? Authors: We thank the reviewer for this suggestion. We have not evaluated that in detail, but that is a very interesting suggestion for future investigations.

The authors have done some interesting analysis already that answers some of the referee's questions but they should go a step further and address how this ultimately would effect a TUG-PDA users emissions output.

Figure 6:
Define what the whiskers (Confidence intervals? What percentage?) are in this box and whisker plot

Line 366:
These laboratory measurements will need to be further defined especially with respect to determining limits of detection for BC as this is something that is not typically done.

Line 378:
This sentence and the paragraph need to be clearer about what differences between the two instruments and their performance are. This sentence I believe is applied to only BC emissions from the AE33 vs from the BCT. The BCT measurements can be separated for the two passing vehicles while the. Measures from the AE33 are not able to be seperated. Is that correct?

Section 3.2.1.
The focus of this section is specifically to compare two BC instruments while the bigger scope of this work is to developing the PDA. I think that this section should be remove or added to the appendix or framed as a case study to emphasize the importance of understand the instruments used with the PDA.

Line 389:
Please edit to be clearer. What is VSP?

Figure 8.
What is roadside?
Also, the fits do not have any statistical information and therefore do not add any meaning takeaways. Edit figure to clearly show the message the authors wish to convey.
Is there a strong statement that supports the placement to be in the middle in order to captures the higher levels of CO2?

Table 3:
Needs to be moved up in the text to where is it referred to. Do not leave it dangling at the end.

---

## Author Response (AR2)

We gratefully thank the reviewer for carefully reading and providing feedback to our manuscript. Below we provide our point-to-point responses to the reviewer's comments. The comments by the reviewer are marked in **black**, responses are marked in **red** and changes to the manuscript are indicated in **blue**.
* * *
Other changes and comments:

- We adapted formula A1 in the Appendix to reflect the calculations used in the TUG-PDA rather than the typical calculation used in the literature. We added the text: "*In our approach (see Eq. A1), we use different start ($t_1$, $t_3$) and stop times ($t_2$, $t_4$) for the pollutant and $CO_2$ integration. Similarly, the BG values are determined independently ($[P]_{tP0}$, $[CO2]_{tCO20}$).*"
- Minor adaptions were done in lines 332, 774-776 and we added acknowledgements.
- If the manuscript is accepted, we would suggest to the editor that the appendices that are not necessary for the understanding of the main part are moved to a supplementary file. This is in our opinion: Appendix B, E, G and H

**Referee 1**

This study developed and demonstrated a point sampling method to automatically measure emissions from a large-scale of individual vehicles. In this works, the authors present their system that can be used for particulate matter (PM) and gas emissions measurements, which is notably independent of vehicle type. They find that when using their peak detection algorithm (TUG-PDA), they can separate vehicle-specific emissions down to a spacing of just a few seconds between vehicles. In this study, they present initial findings from the use of this method that collected ~100,000 vehicle records from several measurement locations, mainly in urban areas. When compared to equivalent remote sensing measurements, the authors found good agreement even with the newest standards which are harder to capture due to their lower emissions and the current remote sensing abilities. This paper is well written and organized.

This manuscript presents novel work on the development of a plume detection system. The authors have done a lot of work to respond to and update their work based on the last round of revisions. With that being said, if the authors are able to update their work with the minor revisions listed in this report, this manuscript should be accepted for publication.
* * *
Line 19: Define NOx here instead of in line 21

Many thanks for this suggestion. $NO_x$ is now introduced in line 19.

*"Of specific interest are nitrogen oxide (NOx) …"*
* * *
Line 47-48: "Other PM metrics such as PN or BC cannot be accurately determined using these systems…"
This point needs to be further clarified for the reader.

We thank the reviewer for this comment. Open-path RES systems only provide PM estimates (PM mass or opacity). They do not measure BC or PN. We revised the description accordingly.

*We replaced "Other PM metrics such as PN or BC cannot be accurately determined using these systems" with "Other PM metrics such as PN or BC are not measured by these systems as they only give PM estimates (Knoll et al., Under review)."*

Table 1 /Table 2: Because Table 2 is not referenced in the text and is just hanging in the section that is placed in, it's recommended to add to Table 1 or putting it in the Appendix and referring to it in the main text.

Table 2 is referenced in the text in line 203: "*The TUG-PDA searches around the vehicle pass time (default window: -1 s to 6 s) for a sequence of positive concentration gradients above a defined threshold (see Table 2)*". Both referees in the previous review asked about the thresholds used for the peak detection, which is also an important information for the reader in our opinion.

*We moved Table 2 directly to 1c (line 207) where it is mentioned to make it clearer to the reader.*

2.2.1 Pre-processing:
This section has a lot of technical information that does not fit well into the bulk of the manuscript. It's recommended to simplify to the following:
1. Raw data from the instruments are time aligned
2. The $CO_2$ is the default time resolution (keep statement about if instruments have large response time differences)
3. Outliers are filtered (state metrics for this) and measurements are smoothed

We are grateful to the reviewer for the suggestion of a simpler and more compact presentation of the pre-processing steps.

*2.2.1 Pre-processing has been revised to (line 175 – 182): "Prior to the actual emissions calculations, three main steps are taken to prepare the raw instrument data.*
1. *Time series data from the different instruments are time-aligned based on manual pollution peaks taken during the measurement campaign (e.g. with a lighter). We align the concentration time series data to the vehicle passes which cause the fastest response (e.g., from vehicle with tailpipe on the same side as the sample extraction).*
2. *The time resolution of the $CO_2$ and pollutant data is equated (default time resolution of 0.5 s) and the $CO_2$ and pollutant data sets are combined into a composite data set.*
3. *The time series data are then smoothed with a rolling Gaussian filter (default window size 5 samples) to reduce the dependence on short variations and outliers. If instruments with large differences in response times ($\Delta t > 2$ s) are used, the response function of the instruments must be aligned."*

Lines 194-201:
This section leading up to describing the algorithm do not add value to the methods section and ends up being more distracting. The audience should understand the concept of PS by this part of the paper and therefore, there is no need to provide these detailed ideas. It's recommended to cut or slim to one introduction statement on why you

developed the algorithm.

We thank the reviewer for this suggestion. From importance is the statement that the $CO_2$ and pollutant emissions are separately processed because this is one of the main differences from previous PS studies.

We shortened the introduction to (line 184): "*We have developed a dedicated algorithm, TUG-PDA, which separates the measured emissions and assigns them to the by-passing vehicles.*"
* * *
Line 211: "There must be either at least two gradients"
Which two gradients are being referred to? As in two pollutants need to be rising and having a gradient? Please clarify in the text.

We wrote (line 208-214): "The TUG-PDA searches around the vehicle pass time (default window: -1 s to 6 s, highlighted in Fig. 4) for a sequence of positive concentration gradients above a defined threshold (see Table 2) of the processed analyte (visualized in Fig. 4). The thresholds were determined based on a large number of manual reviews of TUG-PDA results. There must be either at least two gradients or one very large gradient (> 10 times the threshold) above the threshold."
The thresholds are defined in Table 2.

We shortened and clarified the description and added "defined thresholds" to make it easier for the reader to understand (line 202-205): "*The TUG-PDA searches (default window: -1 s to 6 s) for a sequence of data points with positive concentration gradients above a defined threshold (see Table 2) of the processed analyte around the vehicle pass time (visualized in Fig. 4). There must be either at least two data points of the analyte with a gradient above the threshold or one data point with a very large gradient (> 10 times the threshold).*"
* * *
Figure 3:
This figure can go into the appendix. It is too much to take in and understand in the main text. Also, it's recommended to add step numbers to make it easier to follow the diagram.

We thank the reviewer for this comment.

We added step numbers in the figure and also in the descriptive text. We also restructured the descriptive text (added 1b, 1c and 4) to make it easier to follow and relate it to the figure. We have also clarified in the caption and text that certain processing steps are only carried out for pollutant or $CO_2$ emissions: "*Specific processing steps are only applied to $CO_2$ (3b) or to pollutants (1b, 1e, Stop 4, Stop 5, 3c, 3d, 3.1).*"
As suggested, we moved the detailed flow chart to the appendix.
We drew an overview version of the figure, which we included in the main text, with only the main processing steps to make it easier for the reader to understand (see figure below).

[Figure]

*Figure 3. Emission event processing - flow charts of the peak detection algorithm (TUG-PDA). CO2 and pollutant (e.g., BC, PN, NOx) emissions are processed separately. The algorithm is applied first to CO2 (left) and then to the individual pollutant emissions (right). A detailed flow chart can be found in the Appendix (Fig. C1).*
* * *
Figure 4:
It would be helpful to have the axes go lower than background so the reader can see how the background level is determined and potentially used as a stop (while recognizing that this specific example is not due to background but because of a passing vehicle)

We thank the reviewer for this suggestion.

We adjusted Figure 4 so that the signal is clearly visible down to the background. The background is also subtracted from the areas shown, similar to Figure 5.
* * *
Line 269:
Another recent paper on plume detection was published and should be referenced here.
https://www.sciencedirect.com/science/article/pii/S1364815222003000

We thank the reviewer for providing this interesting paper.

We referred to it in line 262: "*An open source mobile air quality dashboard, including a real-time peak detection algorithm was published by Kelly et al. (2023).*"
* * *
Section 2.2.3:
This section can be revised to be more direct about exactly what this method does. Also, generally, the methods section still seems quite lengthy. It is recommended to highlight the specific and unique points that applies to this PDA system.

We thank the reviewer for this suggestion. We described our methods in detail such that a reader can understand and duplicate our work which was also suggested by reviewer 2 (major revision): "But if the algorithm will not enter the public domain, then I think the manuscript requires major revision to be a more complete methods paper that could be independently duplicated by others."

We revised section 2.2. Data analysis (as suggested by the reviewer, see other comments) to be more understandable and more direct about our method.

We revised section 2.2.3 to be more direct about our method (line 271 to 280): "*Once the ERs of passing vehicles have been determined the measurement results are combined with the vehicle's technical data. Several details from the vehicle technical data are required during the emission analysis to calculate EFs and to perform further statistical analysis. Necessary fields for our post-processing are:*

- *The fuel type (e.g., gasoline, diesel) to calculate fuel-based EFs.*
- *The $CO_2$ emissions measured during the type-approval process of the vehicle model are required to calculate the distance-related EFs.*
- *The European emission standard class is used to classify vehicles according to their emission limits.*
- *The vehicle category is used to perform detailed evaluations for specific vehicle types.*

*With the help of our local partners, we obtained the necessary technical data from the government authorities. The captured license plates are pseudo-anonymized to respect privacy rules.*"
* * *
Lines 306-311:
These sentences make the TUG-PDA sounds like it hypothetically can get down to a detection of 3 s but it is not able to in many cases, which contrasts a lot of the high level take aways from this paper. It is suggested to reword in the more active and present tone to express exactly what the system can do and what factors are able to be adjusted for varying situations.

We thank the reviewer for this suggestion.

We rephrased the text to be specific what the TUG-PDA is capable and what not (line 299-307): "*The TUG-PDA resolves emissions down to a small distance (default: 3 s) between vehicles, if the time between the vehicles is large enough (greater than 3 s) and if a dedicated $CO_2$ peak from the vehicle is observed. Several tests are implemented to determine whether the emissions really come from the current vehicle or are caused by interference from previous vehicles or another source. If other influences are observed, the distance between the vehicles is too small, or overlapping plumes cannot be separated, the measurement is invalid and the emissions for the vehicle cannot be determined. Plume separation can be tuned using several parameters such as gradient thresholds (Table 2), the minimum time allowed between vehicles or the minimum number of samples required as used in the software. This can be very useful for instruments with different response times and for locations with dense traffic to obtain a sufficient number of measurements. Restricting measuring to low-traffic areas would severely limit the application.*"
* * *
Figure 5a:
For V4, the BC integrated areas seems to be cut off early. Though it may be following the rules, the BC plume both starts later and ends later than the corresponding CO2 plume and therefore, should be integrated to basically be a time shifted version of the CO2 plume instead of being cut off which would lead to an underestimation of the EF (this is discussed later on when the author states "The median EFs were 19% lower than in cases without interference"

Because of this, I believe it is very important for the authors to directly respond to the previous comment from Referee 2: Have you characterized how different fractions of peaks captured and assumed baseline concentrations impact resulting emission factors, using the subset of peak events that were 100% isolated with all pollutants starting and ending at the background condition? Authors: We thank the reviewer for this suggestion. We have not evaluated that in detail, but that is a very interesting suggestion for future investigations.

The authors have done some interesting analysis already that answers some of the referee's questions but they should go a step further and address how this ultimately would effect a TUG-PDA users emissions output.

We thank the reviewer for this comment and for pointing this out. For V4, the algorithm underestimates the BC area because the BC background concentration is determined too high. We fully agree that with visual inspection you can see that the BC area is underestimated. It is not so easy to have an automated plume separation that can handle all possible cases of what the emissions look like and accurately calculate the EF. There are some trade-offs or inaccuracies and these will be addressed in the future. We also addressed that in the manuscript between line X1 and X2: *"In the current implementation of the TUG-PDA, the BG determination for overlapping plumes is done by calculating an average value between the median concentration directly between the overlapping plumes and a common BG when no vehicle is passing. This is a simple estimation and entails deviations from the actual situation. This can be seen, for example, in Fig. 5a) for vehicle V4. The BC background is overestimated. This results in a too small integrated area (BC4) and thus underestimated emissions."*

As suggested, we characterized how different fractions of the plume influence the resulting EFs. We added in section 3.1 (line 350 to 354): *"We also looked at how accurate the EF can be calculated using only a fraction of the plume. Therefore, we selected only plumes without interference from other vehicles and calculated EFs using the TUG PDA when the algorithm used only a fraction of the plume in the interval between 3 s and 23 s. Similarly to the investigation shown in Fig. 6, we found that when only a fraction of the plume is used that the EFs are underestimated. The median underestimation for an early cut-off at 3 s is 27 %. The deviation decreases with increasing fraction of the plume (see Appendix Fig. D2)."* We added in Appendix D (line 695 to 699): *"Figure D2 shows how accurate EFs can be calculated when using only a fraction of the plume. Therefore, the algorithm selected 82 plumes that were not affected by emissions from other vehicles. The average plume length of this selection was 18 s and 30 of the plumes were longer than 25 s. The full distribution using the algorithm's defined maximum plume length of 25 s is shown in Figure D2 on the left. Figure D2 (right) shows the deviation from the full plume when only a fraction between 3 s and 23 s is used. The median deviation is maximum at 3 s with 27 % and decreases steadily with increasing plume fraction."*

[Figure]

*Figure D2. Deviation when using only a fraction of the plume to calculate EFs compared to using the entire plume. Left: Distribution of EFs of plumes without interference from other vehicles. Right: Deviation from full plume (25 s) using only fractions between 3 s and 23 s.*

You can also see that the influence on the resulting EFs is relatively small (not included in the manuscript as Figure D2 should contain all this information):

[Figure]

Figure 6:
Define what the whiskers (Confidence intervals? What percentage?) are in this box and whisker plot

We thank the reviewer for pointing this out.

We added in the caption of Figure 6: *"The whiskers represent the 2.5 and the 97.5 percentiles."*

Line 366:
These laboratory measurements will need to be further defined especially with respect to determining limits of detection for BC as this is something that is not typically done.

We thank the reviewer for pointing this out.

We added the following information (line 367): *"We characterized the BCT and the AE33 in the laboratory for properties relevant for PS (see Table 1). A miniCAST soot generator (Jing*

*Ltd, Model 6204 Type B) was used as the particle source. The instruments measured in parallel downstream of a catalytic stripper which removed volatile compounds (Knoll et al., 2021)."*
* * *
Line 378:

This sentence and the paragraph need to be clearer about what differences between the two instruments and their performance are. This sentence I believe is applied to only BC emissions from the AE33 vs from the BCT. The BCT measurements can be separated for the two passing vehicles while the. Measures from the AE33 are not able to be seperated. Is that correct?

We thank the reviewer for this suggestion. Yes this sentence is only applied to the comparison between AE33 and BCT. It is correct that for the shown example the emissions can be separated using the BCT but not using the AE33 because of the slower response time (BCT: 0.9 s, AE33 7 s).

We added at the beginning of the section that the comparison is about BC (line 364): *"For our study we selected two instruments, the custom-designed BCT and the Aethalometer AE33 (Magee Scientific), for their applicability in determining BC emissions using the developed TUG-PDA."*

We clarified what the differences are in the mentioned paragraph (line 380 to 385): *"The emissions captured for the two vehicles overlap, but they can be separated using the BCT. In contrast, the AE33 response time is much slower and the maximum concentration is reached after the second vehicle (V2) has passed by. In this case it is not possible to separate the BC emissions of the two vehicles using the AE33. This example illustrates the importance of choosing instruments with a fast response time when measuring in dense traffic. Individual characteristics (see Table 1), such as the response time, that do not meet the requirements severely limit the application."*
* * *
Section 3.2.1.

The focus of this section is specifically to compare two BC instruments while the bigger scope of this work is to developing the PDA. I think that this section should be remove or added to the appendix or framed as a case study to emphasize the importance of understand the instruments used with the PDA.

We thank the reviewer for this comment. In "3.2 Factors influencing point sampling measurements" we evaluate different influences on PS such as instruments characteristics, measurement location, sampling position and meteorological influences. As the selection of instruments with appropriate characteristics is of great importance in PS, we prefer to leave this section in the main text. If the editor prefers we will move it to the Appendix.

We added at the beginning of Section 3.2.1 that this is about a case study to evaluate instruments for their applicability using the TUG-PDA (line 364): *"For our study we selected two instruments, the custom-designed BCT and the Aethalometer AE33 (Magee Scientific), for their applicability in determining BC emissions using the developed TUG-PDA."*
* * *
Line 389:

Please edit to be clearer. What is VSP?

We introduced VSP in the method section (line X1) and we described it in Appendix G.

We added the reference to the appendix (line 393): "*(VSP, see Appendix G).*"
* * *
Figure 8.
What is roadside?
Also, the fits do not have any statistical information and therefore do not add any meaning takeaways. Edit figure to clearly show the message the authors wish to convey.
Is there a strong statement that supports the placement to be in the middle in order to captures the higher levels of CO2?

The wording "roadside" (roadside measurements) is often used in the literature as synonym for "point sampling" when sampling from the side of the road. Reviewer 2 (major revision) wanted us to show a second trend line for roadside measurements.
* * *
Table 3:
Needs to be moved up in the text to where is it referred to. Do not leave it dangling at the end.

We thank the referee for this suggestion. In the layout of the final paper, we will make sure that the table does not hang at the end.

---

## Author Response (AR3)

We gratefully thank the editor for handling the submission process and for carefully reading and providing feedback to our manuscript. Below we provide our point-to-point responses to the editors's comments. The comments by the editor are marked in **black**, responses are marked in **red** and changes to the manuscript are indicated in **blue**.
* * *
Most comments from the reviewer have been properly addressed. The authors are advised to move part of the technical sections to the Supplement, and to consider improving one of their Figures, as recommended by the referee.
* * *
Additional private note (visible to authors and reviewers only):
Line 19: nitrogen oxides (NOx), with an "s"

We thank the editor for this suggestion.

We adapted the manuscript accordingly
* * *
Line 48: PM mass concentration estimates

We thank the editor for this suggestion.

We changed it to "PM mass emission estimates" as it refers to emission factors.

Since the manuscript is quite long, I suggest to follow the reviewer's suggestion to move as much material as possible in the Supplement.

We agree with the editor. We moved Appendix B, E, F, G and H to the supplementary material. We have left the methods and results section unchanged, as a more detailed description of the methods section provides a detailed insight into the methodology developed, as suggested by a previous reviewer. The results provide a comprehensive overview of the capabilities of the developed method and the factors influencing the measurement approach.

Also please consider the reviewer's second last comment regarding Figure 8, not overlooking the last sentence (see below): "Also, the fits do not have any statistical information and therefore do not add any meaning takeaways. Edit figure to clearly show the message the authors wish to convey. Is there a strong statement that supports the placement to be in the middle in order to captures the higher levels of CO2?"

We thank the editor for pointing this out.

We removed the fits and highlighted regions for measurements from the middle, right and left to emphasize that measurements from the center of the road give significantly higher capture rates (see figure below).

[Figure]